# Targeting local lymphatics to ameliorate heterotopic ossification via FGFR3-BMPR1a pathway

Dali Zhang[1], Junlan Huang[1], Xianding Sun [1,2], Hangang Chen[1], Shuo Huang[1], Jing Yang[1], Xiaolan Du[1], Qiaoyan Tan[1], Fengtao Luo[1], Ruobin Zhang[1], Siru Zhou[1], Wanling Jiang[1], Zhenhong Ni[1], Zuqiang Wang[1], Min Jin[1], Meng Xu[1], Fangfang Li[1], Liang Chen[1], Mi Liu[1], Nan Su[1], Xiaoqing Luo[1], Liangjun Yin[2], Ying Zhu[3], Jerry Q. Feng[4], Di Chen[5], Huabing Qi [1✉], Lin Chen [1✉] & Yangli Xie [1✉]

Acquired heterotopic ossification (HO) is the extraskeletal bone formation after trauma. Various mesenchymal progenitors are reported to participate in ectopic bone formation. Here we induce acquired HO in mice by Achilles tenotomy and observe that conditional knockout (cKO) of *fibroblast growth factor receptor 3* (*FGFR3*) in Col2+ cells promote acquired HO development. Lineage tracing studies reveal that Col2+ cells adopt fate of lymphatic endo-thelial cells (LECs) instead of chondrocytes or osteoblasts during HO development. *FGFR3* cKO in Prox1+ LECs causes even more aggravated HO formation. We further demonstrate that *FGFR3* deficiency in LECs leads to decreased local lymphatic formation in a BMPR1a-pSmad1/5-dependent manner, which exacerbates inflammatory levels in the repaired tendon. Local administration of FGF9 in Matrigel inhibits heterotopic bone formation, which is dependent on FGFR3 expression in LECs. Here we uncover Col2+ lineage cells as an origin of lymphatic endothelium, which regulates local inflammatory microenvironment after trauma and thus influences HO development via FGFR3-BMPR1a pathway. Activation of FGFR3 in LECs may be a therapeutic strategy to inhibit acquired HO formation via increasing local lymphangiogenesis.

---

[1] Department of Wound Repair and Rehabilitation Medicine, State Key Laboratory of Trauma, Burns and Combined Injury, Trauma Center, Research Institute of Surgery, Daping Hospital, Army Medical University, Chongqing, China. [2] Department of Orthopedic Surgery, The Second Affiliated Hospital, Chongqing Medical University, Chongqing, China. [3] Department of Rehabilitation Medicine, The First Affiliated Hospital, Chongqing Medical University, Chongqing, China. [4] Texas A&M University College of Dentistry, Dallas, TX, USA. [5] Research Center for Human Tissues and Organs Degeneration, Shenzhen Institutes of Advanced Technology, Chinese Academy of Sciences, Shenzhen, China. ✉email: hbqi3@163.com; linchen70@163.com; xieyangli841015@163.com

Heterotopic ossification (HO) is the ectopic bone formation in soft tissues. Hereditary HO includes fibrodysplasia ossificans progressive (FOP) and progressive osseous heteroplasia (POH). FOP is mostly caused by heterozygous gain-of-function mutation (R206H) in *ACVR1/ALK2* and is usually initiated in childhood through endochondral ossification[1]. POH is intramembranous ectopic bone formation caused by inactivation mutations of *GNAS*[2]. Both genetic diseases develop progressively with no effective solution.

Acquired HO usually develops after major trauma, burn, surgery, or central nervous system injury with a high morbidity[3]. Acquired HO is found in about 30% cases of orthopedic surgery such as hip arthroplasty. HO rate rises to above 60% in war-wounded patients and above 90% in patients with severe traumatic amputations[4,5]. Patients with HO often suffer from chronic pain, joint contractures, and limited mobility, which further incapacitates normal gait and daily activities. Currently, clinical therapy of acquired HO is limited to radiation, NSAIDs, and surgery with high recurrence rates[4].

Acquired HO has been reported to develop through endochondral ossification involving stages of inflammation, chondrogenesis, osteogenesis, and maturation. Immune cells and osteogenic progenitors are involved in HO development and various growth factors such as TGF-β have been reported to play important roles in regulating HO formation[3]. HO development requires the coordinated effects of three essential factors: osteogenic progenitor cells, molecular signals triggering ectopic bone formation, and proper microenvironment[6]. Various osteogenic progenitors including fibroblasts[7], endothelial cells[8], and mesenchymal stem cells (MSCs) that are positive for Prx1[9], Gli1[10], and nestin[3] have been reported to participate in HO formation through diverse signaling pathways. Osteogenic signaling pathways such as BMP signaling play central roles in HO development. *ACVR1* mutation leads to FOP by over-activating BMP signaling and inhibition of BMP type I receptor reduces ectopic bone formation in a mouse FOP model[11]. Local BMP2 application has been used to induce acquired HO formation[3]. Inflammation following trauma has been known to trigger ectopic bone formation. Multiple immune cells including macrophages, mast cells, and lymphocytes, as well as various inflammatory cytokines including TNF-α and IL-1β are involved in HO development[12–18]. Thus, reducing inflammatory levels with NSAIDs, etc. has been widely accepted as a practical strategy for HO prophylaxis[4,18].

Lymphatic vessels play an important role in the clearance of local inflammatory cells and cytokines[19,20]. Vascular endothelial growth factor C (VEGF-c) promotes lymphatic growth by activating VEGF receptor 3 (VEGFR3) in lymphatic endothelium[21]. VEGF-c treatment stimulates cardiac lymphangiogenesis and inflammation resolution after myocardial infarction[22], leading to improved cardiac function[23]. Targeted delivery of VEGF-c mediated by fusion proteins was reported to induce local lymphangiogenesis and reduce chronic skin inflammation[21]. Injection of adeno-associated virus expressing VEGF-c promotes lymphatic vessels and attenuates joint damage in a mouse model of rheumatoid arthritis[24]. Achilles tenotomy has been widely used to induce acquired HO formation in the Achilles tendon of mice. Lymphatics are absent in the normal Achilles tendon, but are detectable in the peritendineum and musculo-tendineal transition zone. Interestingly, lymphatic vessels are detected in the rearranged Achilles tendon 2 weeks after surgery in a rat tendon lesion model[25]. The detailed role of lymphatics in HO development remains unclarified.

Fibroblast growth factor (FGF) signaling plays an essential role in skeletal development[26]. Activation mutations of *fibroblast growth factor receptor 3* (*FGFR3*) in human cause chondrodysplasia including achondroplasia, hypochondroplasia as well as thanatophoric dysplasia through inhibiting chondrocyte proliferation and differentiation[27]. Meanwhile, FGFR3 also plays a vital role in the regulation of lymphatic formation. FGFR3 is expressed in human and mouse lymphatic endothelial cells (LECs) and is essential for LEC proliferation and migration[28]. 9-cis retinoic acid (9-cisRA) promotes LEC proliferation, migration, and tube formation via activating FGF signaling. 9-cisRA-induced proliferation of LECs is coupled with increased FGFR3 expression, which is suppressed by soluble FGFR3 recombinant protein that sequesters FGF ligands[29]. All these findings suggest the possible involvement of FGFR3 in acquired HO development, although the accurate role and detailed underlying mechanisms remain to be clarified.

In this study, we explored whether FGFR3, an important regulator of endochondral ossification and lymphatic formation, influences HO development. MSCs labeled by Prx1[9], Gli1[10], and nestin[3], etc. have been found to contribute to acquired HO formation in diverse ways. Collagen 2 (Col2) is a specific molecule for chondrocytes, which highly express FGFR3[26]. Meanwhile, Col2+ cells were reported to function as mesenchymal progenitors in mice during postnatal life in recent years[30–32]. However, the role of Col2+ mesenchymal progenitors and the involvement of FGFR3 in HO formation remain unclear. Therefore, we conditionally deleted *FGFR3* in Col2+ cells (*FGFR3^Col2*) and found that acquired HO progression was aggravated in Achilles tenotomy model of *FGFR3^Col2* mice. Unexpectedly, lineage tracing revealed that Col2+ cells adopted fate of LECs instead of chondrocytes or osteoblasts in the Achilles tendon after surgery in both *Col2-CreER^T2; R26R^tdTomato* and *FGFR3^f/f; Col2-CreER^T2; R26R^tdTomato* mice as well as in *Col2-CreER^T2; R26R^mTmG* mice. The number of Col2+ cells-derived lymphatics was significantly decreased in mice with *FGFR3* deficiency. Deletion of *FGFR3* in LECs inhibited lymphatic formation via upregulated BMPR1a-pSmad1/5 signaling, which decreased lymphatic drainage in the repaired Achilles tendon with resultantly increased local inflammatory levels, leading to aggravated HO development.

## Results

**FGFR3 deficiency in Col2+ cells promotes acquired HO development.** To test whether FGFR3 influences acquired HO formation after trauma, we bred *FGFR3^f/f* mice to *Col2-CreER^T2* mice expressing a tamoxifen-inducible Cre recombinase in Col2+ cells, which have been reported to encompass early mesenchymal progenitors[30,31], to obtain *FGFR3^f/f; Col2-CreER^T2* mice (*FGFR3^Col2*). We performed Achilles tenotomy to induce acquired HO in 10-week-old *FGFR3^Col2* mice after 5 consecutive days of tamoxifen administration (Fig. 1a). X-ray analysis of mice at 2, 4, 6, and 8 weeks post surgery demonstrated that the HO incidence was higher in *FGFR3^Col2* mice compared with that of *FGFR3^f/f* mice (Supplementary Table 1). 18.2% (4/22) *FGFR3^Col2* mice were observed to develop acquired HO at 2 weeks post tenotomy relative to 4.3% (1/23) in control mice. At 8 weeks post surgery, 95.5% (21/22) *FGFR3^Col2* mice developed evident HO in the Achilles tendon compared with 78.3% (18/23) in *FGFR3^f/f* controls (Supplementary Table 1). Meanwhile, X-ray and μCT analysis revealed enlarged heterotopic bone in the Achilles tendon of *FGFR3^Col2* mice relative to that in *FGFR3^f/f* controls at 4 weeks after tenotomy (Supplementary Fig. 1a). H&E and Safranin O/Fast Green (SOFG) staining revealed abundant cartilage in the repaired tendon of *FGFR3^Col2* mice at 4 weeks after Achilles tenotomy, whereas few chondrocytes were observed in the tendon of control mice (Supplementary Fig. 1b-d). Immunostaining showed that the numbers of Sox9+ chondrocytes and Osx+ osteoblasts were significantly increased in the ectopic bone in

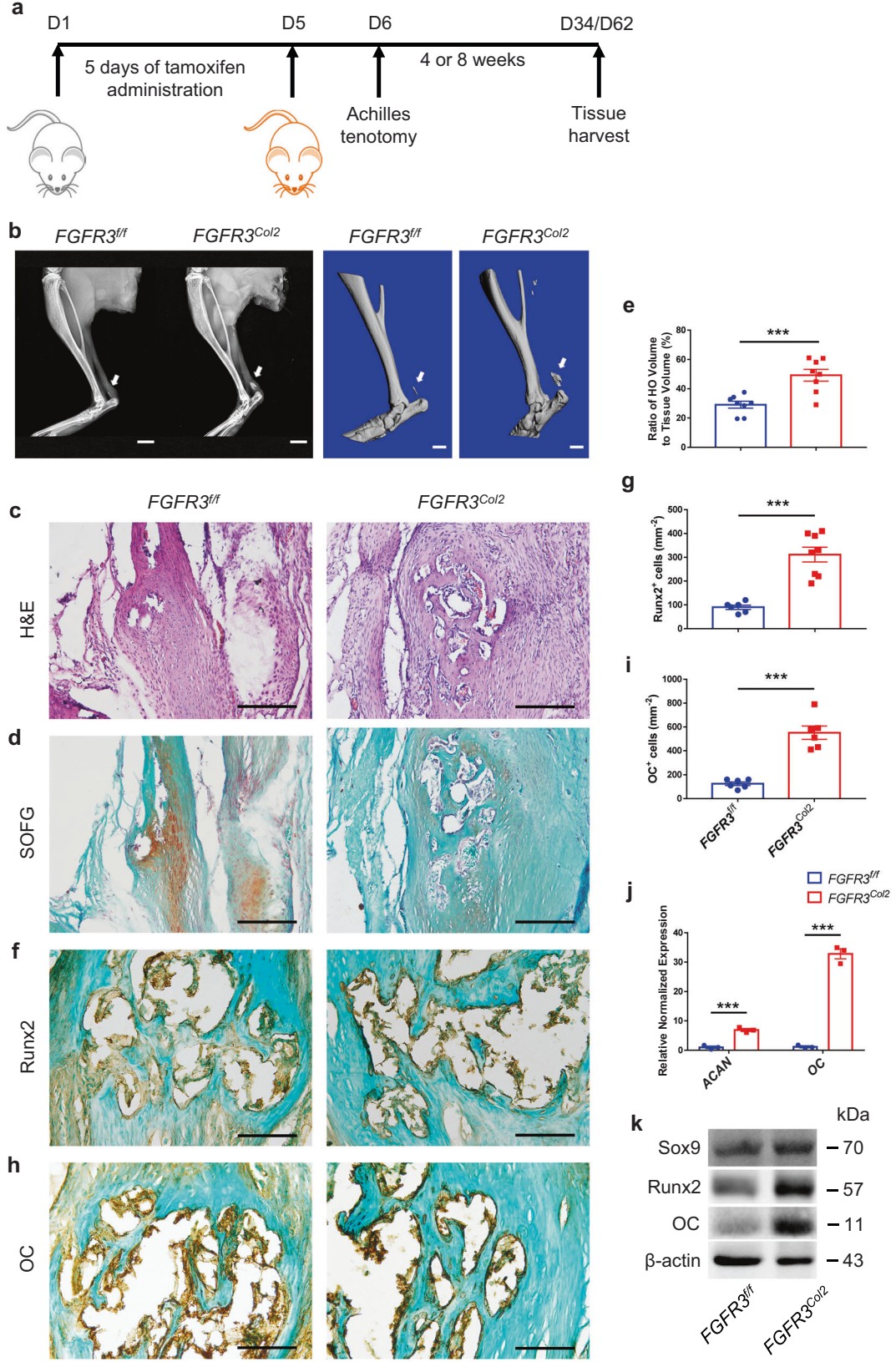

*FGFR3^{Col2}* mice relative to that of the controls (Supplementary Fig. 1e-h). Furthermore, quantitative PCR and western blot demonstrated that the expressions of chondrogenic markers aggrecan (ACAN) and Sox9, as well as the osteogenic marker Cbfa1/Runx2 in the Achilles tendon were significantly higher in

*FGFR3^{Col2}* mice 4 weeks after surgery (Supplementary Fig. 1i, j). At 8 weeks post surgery, significantly increased HO formation with fully developed cancellous bone and marrow was observed in the repaired tendon of *FGFR3^{Col2}* mice relative to *FGFR3^{f/f}* controls (Fig. 1b–e). Moreover, immunostaining demonstrated

**Fig. 1 FGFR3 deficiency in Col2+ lineage cells promotes acquired HO formation. a** Experimental strategy of tamoxifen administration and acquired HO induction. **b** Representative X-ray (left) and μCT (right) images of ectopic bone in *FGFR3^f/f^* and *FGFR3^Col2^* mice at 8 weeks after tenotomy. n = 8 per group. Scale bars, 2 mm for X-ray; 1 mm for μCT. **c–e** Representative H&E and SOFG images of ectopic bone in *FGFR3^f/f^* and *FGFR3^Col2^* mice at 8 weeks after tenotomy and histomorphometry analysis. n = 8 per group. Scale bars, 200 μm. **f–i** Representative immunohistochemical staining of Runx2 (**g** *FGFR3^f/f^* = 6, *FGFR3^Col2^* = 8) and OC (**i** *FGFR3^f/f^* = 6, *FGFR3^Col2^* = 6) and relative quantitative analysis (**g, i**). Scale bars, 100 μm. **j** mRNA levels of *ACAN* and *OC* in the Achilles tendon of *FGFR3^f/f^* and *FGFR3^Col2^* mice at 8 weeks post surgery. n = 3 per group. **k** Protein levels of Sox9, Runx2, and OC in the Achilles tendon of *FGFR3^f/f^* and *FGFR3^Col2^* mice at 8 weeks post surgery. n = 3 per group. All data represent mean ± SEM. ***P < 0.001 by unpaired two-tailed Student's *t*-test.

that the numbers of Runx2+ and osteocalcin+ (OC+) osteoblasts in HO lesions were significantly increased in *FGFR3^Col2^* mice compared with controls (Fig. 1f–i). Furthermore, quantitative PCR and western blot analysis revealed significantly increased expressions of ACAN, Sox9, Runx2, and OC in the tendon of *FGFR3^Col2^* mice, which confirmed aggravated heterotopic bone formation in mice with *FGFR3* deficiency (Fig. 1j, k). Altogether, our data revealed that conditional deletion of *FGFR3* in Col2+ lineage cells aggravates the initiation and progression of acquired HO.

**Col2+ cells adopt LEC fate during acquired HO formation.** To clarify the contribution of Col2+ lineage cells during HO development, lineage tracing of Col2+ cells was performed using tamoxifen-inducible *Col2-CreER^T2^; R26R^tdTomato^* mice (*Col2^tomato^*) and *FGFR3^f/f^; Col2-CreER^T2^; R26R^tdTomato^* mice (*FGFR3^f/f^-Col2^tomato^*). Tamoxifen was administered for 5 consecutive days to label Col2+ cells, followed by Achilles tenotomy to induce acquired HO initiation. Consistent with previous reports[30,31], tdTomato-labeled cells were detectable in the bone marrow and cortical bone of *Col2^tomato^* mice at 8 weeks after tamoxifen induction. We also observed that tdTomato-labeled Col2+ lineage cells contributed to the muscles of paws in mice (Supplementary Fig. 2a). Unexpectedly, tdTomato-labeled Col2+ lineage cells were not observed in HO lesions with substantial expressions of Sox9 and Runx2 in *Col2^tomato^* and *FGFR3^f/f^-Col2^tomato^* mice at 8 weeks post surgery. Moreover, tdTomato-labeled Col2+ lineage cells were found distributed in linear patterns along the repaired Achilles tendons (Fig. 2a–d). Considering the abundant angiogenesis observed in Achilles tendons after surgery (Fig. 2e), we speculated that Col2+ lineage cells might participate in the formation of blood vessels in repaired Achilles tendons. However, immunostaining for CD31 in both *Col2^tomato^* and *FGFR3^f/f^-Col2^tomato^* mice revealed that Col2+ cells were not involved in angiogenesis in the tendon after trauma, though tdTomato-labeled cells also expressed CD31 (Fig. 2f). Previous findings demonstrated that newly formed lymphatic vessels were detected in the repaired Achilles tendons after surgery[25] and LECs also expressed CD31[25,33,34]. Thus, we further investigated whether these linearly distributed Col2+ lineage cells might be associated with newly formed lymphatics in the Achilles tendon post surgery. Immunostaining revealed abundant expressions of canonical LEC marker LYVE1 in tdTomato-labeled Col2+ lineage cells in the repaired tendon of *Col2^tomato^* mice at 8 weeks post Achilles tenotomy (Fig. 2g). Meanwhile, linearly distributed Col2+ lineage cells in the connective tissues near the Achilles tendon were also observed to express LYVE1 (Fig. 2h). Immunostaining for LYVE1 demonstrated that Col2+ lineage cells in the peritendineum of uninjured Achilles tendon adopted LEC fate at 8 weeks after tamoxifen induction (Fig. 2i). Considering *mTmG* reporter is less susceptible to basal *CreER^T2^* leakage[35], *Col2-CreER^T2^; R26R^mTmG^* mice (*Col2^mTmG^*) were also used to confirm the LEC identity of Col2-derived cells in the tendon after injury. Col2+ lineage cells in the repaired tendon of *Col2^mTmG^* mice 4 weeks post tenotomy expressed LYVE1, Prox1, which is the specific marker of LECs, as

well as PDPN (Fig. 2j–l). Tamoxifen-independent CreER^T2^ activity was tested in control mice including *Col2^tomato^* and *tomato* mice as well as *Col2^mTmG^* and *mTmG* mice without tamoxifen induction. We did not find reporter-positive cell in the repaired Achilles tendon of these control mice at 4 weeks post surgery, though co-staining of LYVE1 revealed that lymphatics were already formed in these tendons (Supplementary Fig. 3a, b). In vitro evidence of primary cells isolated from repaired Achilles tendons of *Col2^mTmG^* mice at 4 weeks post surgery showed more evident co-stainings of LEC markers including LYVE1, Prox1, and VEGFR3 with GFP-labeled Col2+ lineage cells (Supplementary Fig. 4a). Tamoxifen-independent reporter activation was not observed in the isolated primary cells of repaired tendons of *mTmG* mice and *Col2^mTmG^* mice without tamoxifen induction at 4 weeks post surgery (Supplementary Fig. 4b). It was reported that LYVE1 also labels macrophages[36]. Therefore, we conducted co-immunostaining of LYVE1 and F4/80 and demonstrated that tdTomato-labeled cells in the tendon of *Col2^tomato^* and *FGFR3^f/f^-Col2^tomato^* mice 8 weeks post surgery expressed LYVE1 instead of F4/80, which further confirmed the LEC identity rather than macrophage fate of Col2-derived cells in the tendon post surgery (Supplementary Fig. 5a, b).

Since Col2+ cells could serve as mesenchymal progenitors as previously reported[30–32], we then generated *Prx1-CreER^T2^; R26R^tdTomato^* (*Prx1^tomato^*) and *FGFR3^f/f^; Prx1-CreER^T2^; R26R^tdTomato^* mice (*FGFR3^f/f^-Prx1^tomato^*) to determine whether paired related homeobox 1+ (Prx1+) mesenchymal progenitors might also participate in lymphatic formation in the tendon after Achilles tenotomy. Interestingly, immunostaining for Sox9 and LYVE1 demonstrated that Prx1+ lineage cells contributed to acquired HO formation as reported in previous study[9], but failed to adopt LEC fate in the Achilles tendon 8 weeks after surgery (Supplementary Fig. 6a, b). These results indicate that Col2+ lineage cells play a distinct role in HO development by contributing to local lymphatic formation instead of ectopic chondrogenesis or osteogenesis.

Previous studies revealed that lymphatic vessels were absent in normal Achilles tendons[25]. Consistently, LYVE1+ LECs and Col2+ lineage cells were not observed in the Achilles tendon without tenotomy. Instead, tdTomato-labeled Col2+ cells were detected in the peritendineum (Fig. 2m). Eight weeks after tenotomy, abundant Col2+ lineage cells were observed to express LYVE1 as well as the mesenchymal marker α-SMA in the repaired Achilles tendon and peritendineum of *Col2^tomato^* mice (Fig. 2n and Supplementary Fig. 2b). Altogether, we speculate that Col2+ cells in the peritendineum and adjacent connective tissues act as LEC progenitors contributing to lymphatics in the Achilles tendon post surgery.

**FGFR3 deficiency in LECs aggravates acquired HO formation.** We next examined the effect of *FGFR3* deficiency in LECs on HO development. Prospero homeobox protein 1 (Prox1) has been confirmed as an essential transcription factor for LEC differentiation[28]. *FGFR3^f/f^; Prox1-CreER^T2^* mice (*FGFR3^Prox1^*) were used to reveal the role of FGFR3 in LECs during HO

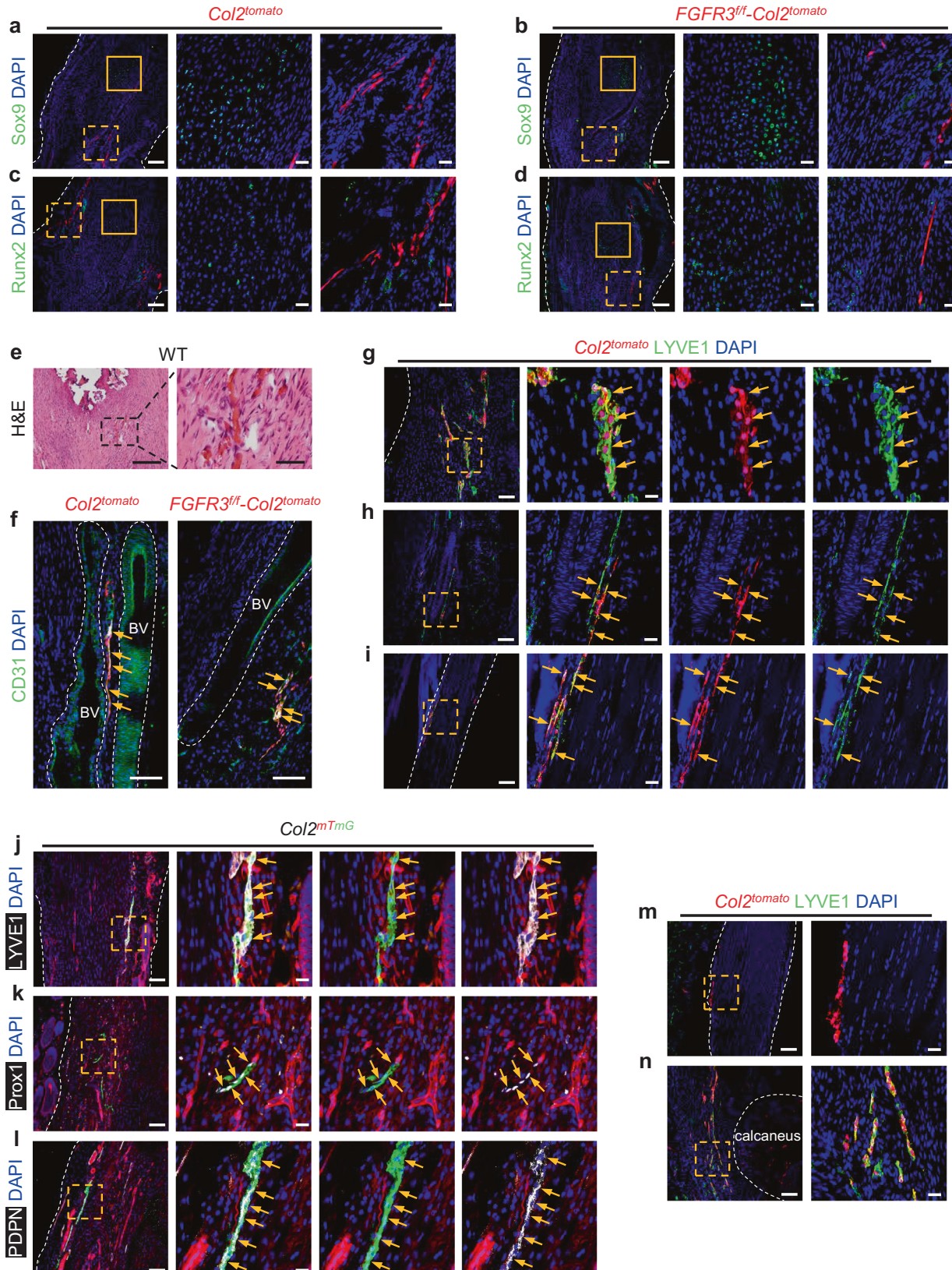

pathogenesis. After 5 days of tamoxifen treatment, the mice underwent Achilles tenotomy to induce acquired HO. Similar to *FGFR3^Col2* mice, HO volume in the Achilles tendon was significantly increased in *FGFR3^Prox1* mice compared with that in controls 4 weeks post surgery as indicated by X-ray and μCT analysis (Fig. 3a, b). H&E and SOFG staining showed very few

numbers of chondrocytes in the tendon of control mice, which were significantly increased in *FGFR3^Prox1* mice (Fig. 3c–e). Immunohistochemical (IHC) data revealed significantly higher expressions of Sox9 and Osx in the HO lesions of *FGFR3^Prox1* mice relative to *FGFR3^f/f* mice (Fig. 3f–i). Significantly increased HO volume was detected in *FGFR3^Prox1* mice relative to controls

**Fig. 2 Col2$^+$ lineage cells adopt LEC fate in the Achilles tendon of mice after tenotomy. a–d** Representative confocal images of ectopic bone sections stained with Sox9 (**a**, **b**) or Runx2 (**c**, **d**) (green) and DAPI (blue) in *Col2$^{tomato}$* (**a**, **c**) and *FGFR3$^{f/f}$-Col2$^{tomato}$* mice (**b**, **d**) at 8 weeks after Achilles tenotomy. $n = 5$ per group. Solid line boxes indicate HO lesion (higher magnification, middle). Dashed line boxes indicate the site of tdTomato-labeled cells (higher magnification, right). Scale bars, 100 μm (left); 20 μm (middle and right). **e** Representative H&E images of repaired Achilles tendon in wild-type (WT) mice ($n = 7$) at 8 weeks after surgery. Dashed line boxes indicate the site of newly formed blood vessels (higher magnification, right). Scale bars, 200 μm (left); 50 μm (right). **f** Representative confocal images of CD31 (green) and DAPI (blue) immunostained Achilles tendon sections in *Col2$^{tomato}$* (left) and *FGFR3$^{f/f}$-Col2$^{tomato}$* mice (right) at 8 weeks after surgery. $n = 5$ per group. Blood vessels (BV) are marked with dashed lines. Scale bars, 100 μm. **g–i** Representative confocal images of sections of repaired Achilles tendon (**g**), connective tissue near the Achilles tendon (**h**), and uninjured Achilles tendon (**i**) immunostained with LYVE1 (green) and DAPI (blue) in *Col2$^{tomato}$* mice at 8 weeks after surgery. $n = 4$–6 per group. Dashed line boxes indicate LYVE1$^+$ tdTomato-labeled cells (higher magnification with split channels, right). Scale bars, 100 μm (left); 20 μm (right). **j–l** Representative confocal images of repaired Achilles tendon sections stained with LYVE1 (**j**), Prox1 (**k**) or PDPN (**l**) (white), and DAPI (blue) in *Col2$^{mTmG}$* mice at 4 weeks after tamoxifen injection. $n = 4$ per group. Dashed line boxes indicate the site of GFP-labeled cells (higher magnification with split channels, right). Scale bars, 100 μm (left); 20 μm (right). **m**, **n** Representative confocal images of uninjured Achilles tendon in *Col2$^{tomato}$* mice 1 day after tamoxifen induction (**m**) and repaired Achilles tendon with peritendineum and adjacent connective tissues in *Col2$^{tomato}$* mice at 8 weeks after surgery (**n**) immunostained with LYVE1 (green) and DAPI (blue). $n = 4$ per group. Dashed line boxes indicate the site of tdTomato-labeled cells (higher magnification, right). Bone tissue of calcaneus is marked with dashed lines. Scale bars, 100 μm (left); 20 μm (right). Yellow arrows indicate Col2-derived cells labeled by CD31 (**f**), LYVE1 (**g–j**), Prox1 (**k**), or PDPN (**l**). White dashed lines indicate outlines of the tendon (**a–d**, **g**, **i–m**).

at 8 weeks post Achilles tenotomy (Fig. 3j, k). Abundant well-developed cancellous bone and ectopic cartilage were observed in *FGFR3$^{Prox1}$* mice. By contrast, thinner cartilage layers with limited cancellous bone were detected in control mice (Fig. 3l–n). Runx2 and OC expressions in the tendon were increased in *FGFR3$^{Prox1}$* mice compared with controls (Fig. 3o–r). Collectively, *FGFR3* cKO in LECs tremendously promoted acquired HO development, which indicates that there appears to be a causal relationship between dysregulated lymphangiogenesis and HO development represented in both *FGFR3$^{Prox1}$* and *FGFR3$^{Col2}$* mouse models.

***FGFR3* deficiency in LECs inhibits local lymphatic formation and enhances inflammation in the Achilles tendon after trauma.** Previous studies have demonstrated that FGFR3 is essential for the proliferation and migration of LECs[28,29]. We examined local lymphatic vessels in the tendon of *FGFR3* cKO mice post surgery. Immunostaining for LYVE1 demonstrated that about 82.9 ± 5.2% of tdTomato-labeled Col2$^+$ lineage cells in the tendon adopted LEC fate in *Col2$^{tomato}$* mice at 4 weeks post Achilles tenotomy, which was increased to 99.1 ± 0.9% at 8 weeks after surgery. However, fewer Col2$^+$ lineage cells adopted LEC fate at 4 and 8 weeks post surgery (67.8 ± 3.8% and 83.3 ± 6.0%, respectively) in *FGFR3$^{f/f}$-Col2$^{tomato}$* mice relative to *Col2$^{tomato}$* mice (Fig. 4a–c). TdTomato-labeled Col2$^+$ lineage cells contributed to approximately 53.3 ± 3.4% and 56.6 ± 2.8% of LYVE1$^+$ LECs in the tendon of *Col2$^{tomato}$* mice at 4 and 8 weeks post surgery, respectively, which were reduced to 43.2 ± 6.2% and 45.6 ± 4.9% in *FGFR3$^{f/f}$-Col2$^{tomato}$* mice (Fig. 4d). Lymphatics derived from Col2$^+$ lineage cells were significantly decreased in the repaired tendon of *FGFR3$^{f/f}$-Col2$^{tomato}$* mice relative to controls (Fig. 4e). Furthermore, quantitative PCR showed that the expressions of LEC markers including *LYVE1*, *Prox1*, and *PDPN* in repaired Achilles tendons of *FGFR3$^{Col2}$* mice were significantly reduced relative to *FGFR3$^{f/f}$* mice (Fig. 4f). Moreover, immunostaining of LYVE1 revealed that lymphatic formation in the tendon was also significantly inhibited in *FGFR3$^{Prox1}$* mice compared with that of controls at 8 weeks post tenotomy (Fig. 4g–i). In addition, the number of LYVE1$^+$ LECs was not affected in the repaired tendon of *FGFR3$^{f/f}$-Prx1$^{tomato}$* mice compared with *Prx1$^{tomato}$* mice (Supplementary Fig. 6b, c). Taken together, *FGFR3* deficiency in LECs inhibits local lymphatic formation in the Achilles tendon after tenotomy.

Lymphatics serve as the drainage system to remove local interstitial fluid, inflammatory cytokines, and cells[19,20]. We

further explored whether *FGFR3* deficiency-related decreased lymphatic formation influenced the local lymphatic drainage and inflammatory status in Achilles tendons after tenotomy. Indocyanine green (ICG) near-infrared (NIR) imaging was performed to determine the lymphatic draining function in repaired Achilles tendons as previously described[37]. Significantly decreased ICG clearance was detected in *FGFR3$^{Col2}$* and *FGFR3$^{Prox1}$* mice relative to controls 24 h after ICG injection in the footpad, indicating impaired local lymph drainage in mice with *FGFR3* deficiency (Fig. 4j, k). Previous studies demonstrated that lymphatics are critical for the resolution of local tissue inflammation including removal of inflammatory macrophages[19] that may trigger HO formation by producing TGF-β1[3,38], oncostatin M[13], and BMP4[14], etc. Four weeks post surgery, significantly increased numbers of inflammatory M1 macrophages expressing F4/80 and iNOS were observed in repaired tendons of *FGFR3* cKO mice relative to controls (Fig. 4l, m). Eight weeks post trauma, abundant M1 macrophages remained in the Achilles tendon of mutant mice, which were almost completely resolved in control mice (Fig. 4l, n). These data suggest that sustained high-level inflammation after trauma is related to impaired local lymphatic drainage in *FGFR3*-deficient mice, which may aggravate HO development.

Next, we examined the pathogenesis of acquired HO in human specimens. SOFG staining of HO specimens at osteogenesis stage (3–6 months after trauma) showed thick cartilage layers adjacent to limited cancellous bone. By contrast, during maturation stage (12–18 months after trauma), abundant well-developed cancellous bone with thinner cartilage was observed (Supplementary Fig. 7a). To determine local lymphatic formation during human HO progression, we performed immunofluorescence (IF) of LYVE1 and FGFR3 on human HO specimens. Abundant LYVE1$^+$ lymphatics with high level of FGFR3 expression (61.7 ± 6.5%) were observed in HO lesions at osteogenesis stage from patients. In contrast, a significant reduction of LYVE1$^+$ lymphatics with remarkably decreased FGFR3 expression (10.5 ± 1.9%) was observed in HO lesions during maturation stage (Supplementary Fig. 7b-d). Meanwhile, immunostaining revealed a significantly increased number of F4/80$^+$ macrophages in HO lesions at maturation stage relative to osteogenesis stage and the percentage of F4/80$^+$iNOS$^+$ inflammatory macrophages was also increased though without significant difference (Supplementary Fig. 7e, f). During acquired HO progression in mice, local lymphatic vessels were significantly reduced with strongly elevated numbers of inflammatory macrophages in the Achilles tendon 12 weeks after surgery compared with that of 8 weeks post tenotomy

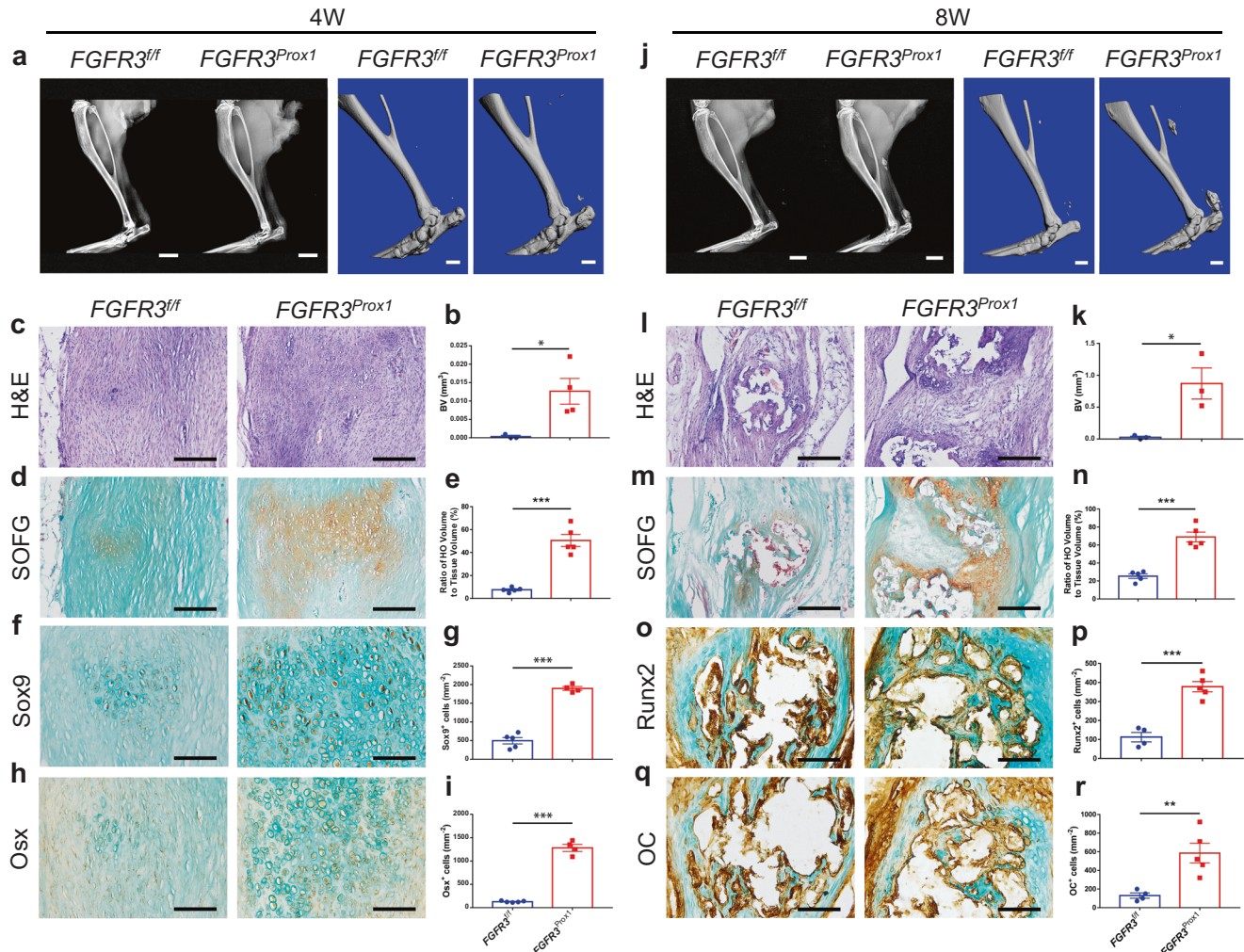

**Fig. 3 *FGFR3* deletion in LECs aggravates heterotopic bone formation. a, b** Representative X-ray (left) and µCT (right) images of heterotopic bone in the Achilles tendon and quantitative analysis in *FGFR3^{f/f}* (n = 3) and *FGFR3^{Prox1}* mice (n = 4) at 4 weeks after tenotomy. Scale bars, 2 mm for X-ray; 1 mm for µCT. **c–e** Representative H&E and SOFG images of heterotopic bone and histomorphometry analysis in *FGFR3^{f/f}* and *FGFR3^{Prox1}* mice at 4 weeks after surgery. n = 5 per group. Scale bars, 200 µm. **f–i** Representative immunohistochemical (IHC) staining of Sox9 (**f** *FGFR3^{f/f}* = 5, *FGFR3^{Prox1}* = 4) and Osx (**h** *FGFR3^{f/f}* = 5, *FGFR3^{Prox1}* = 4) and relative quantitative analysis (**g, i**). Scale bars, 100 µm. **j, k** Representative X-ray (left) and µCT (right) images of ectopic bone in the tendon and quantitative analysis (**k**) in *FGFR3^{f/f}* (n = 3) and *FGFR3^{Prox1}* mice (n = 3) at 8 weeks after Achilles tenotomy. Scale bars, 2 mm for X-ray; 1 mm for µCT. **l–n** Representative H&E and SOFG images of ectopic bone and histomorphometry analysis in *FGFR3^{f/f}* and *FGFR3^{Prox1}* mice at 8 weeks after tenotomy. n = 5 per group. Scale bars, 200 µm. **o–r** Representative immunohistochemical staining of Runx2 (**o** *FGFR3^{f/f}* = 4, *FGFR3^{Prox1}* = 5) and OC (**q** *FGFR3^{f/f}* = 4, *FGFR3^{Prox1}* = 5) and relative quantitative analysis (**p, r**). Scale bars, 100 µm. All data represent mean ± SEM. *P < 0.05; **P < 0.01; ***P < 0.001 by unpaired two-tailed Student's t-test.

(Supplementary Fig. 7g-j). Taken together, our data from mice and human samples suggest that FGFR3 is essential for the local lymphatic formation during post-traumatic HO formation and downregulated FGFR3 expression of LECs in the inflammatory milieu during HO progression leads to decreased lymphatic formation, which further aggravates inflammation levels promoting HO development.

**BMPR1a-pSmad1/5 signaling acts downstream of FGFR3 in LECs to inhibit local lymphatic formation and aggravate acquired HO development.** BMP2-pSmad1/5/8 signaling plays a negative role in lymphatic development via attenuating Prox1 expression and BMP signaling activity is attenuated in developing LECs[39]. We have previously demonstrated that FGFR3 inhibits BMPR1a-pSmad1/5 signaling in chondrocytes and conditional deletion of *FGFR3* in chondrocytes upregulates BMPR1a-pSmad1/5 levels[27]. To examine whether *FGFR3* deficiency inhibits lymphatic formation in the traumatically injured Achilles

tendon by upregulating BMPR1a-pSmad1/5 signaling in LECs, we examined the expressions of BMPR1a and pSmad1/5 in the tendon of *Col2^{tomato}* and *FGFR3^{f/f}-Col2^{tomato}* mice at 8 weeks post surgery. Immunostaining revealed that FGFR3 expression in tdTomato-labeled LYVE1^+ LECs was significantly decreased in the repaired tendon of *FGFR3^{f/f}-Col2^{tomato}* mice (11.4 ± 3.0%) compared with controls (77.4 ± 5.6%) 8 weeks post tenotomy (Fig. 5a, b). Co-immunostaining of LYVE1 and BMPR1a revealed that tdTomato-labeled Col2^+ cells adopted LEC fate with 9.0 ± 0.7% BMPR1a expression in repaired Achilles tendons of *Col2-tomato* mice. By contrast, significantly increased BMPR1a expression (40.1 ± 5.4%) was observed in tdTomato-labeled LYVE1^+ LECs in the tendon of *FGFR3^{f/f}-Col2^{tomato}* mice (Fig. 5c, d). Similar results were found by co-immunostaining of LYVE1 and pSmad1/5. The expressions of pSmad1/5 in tdTomato-labeled LYVE1^+ LECs were significantly increased in *FGFR3^{f/f}-Col2^{tomato}* mice (48.4 ± 5.6%) relative to *Col2^{tomato}* mice (10.6 ± 1.8%) (Fig. 5e, f). Furthermore, *FGFR3* knockdown in a

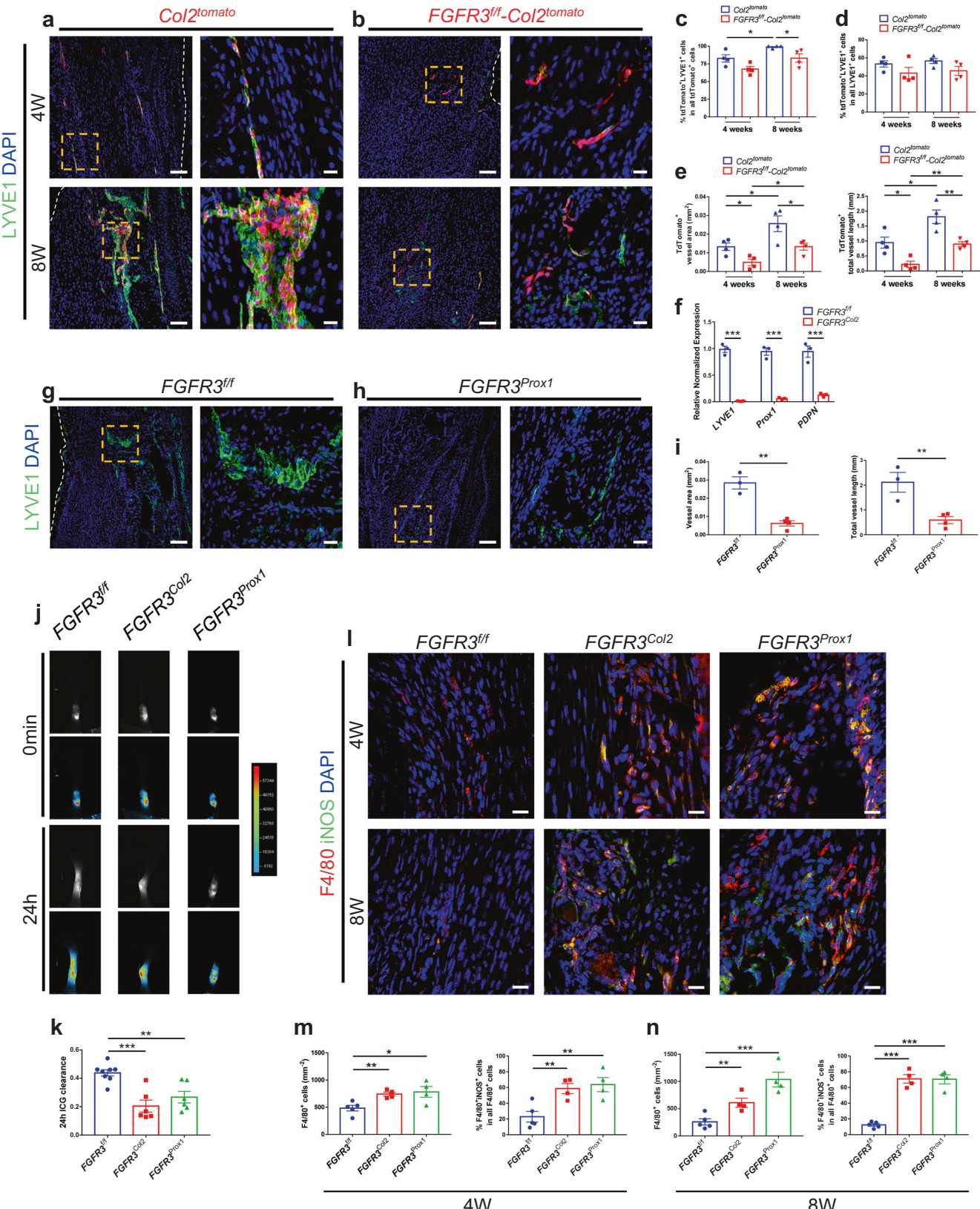

mouse LEC cell line led to upregulated BMPR1a and pSmad1/5 expressions without altered Smad1/5 expression compared with control (CON) (Fig. 5g). Consistently, immunostaining demonstrated that FGFR3 expression in LYVE1+ LECs was remarkably decreased in the repaired tendon of *FGFR3^Prox1* mice compared

with controls at 8 weeks post surgery (Supplementary Fig. 8a). BMPR1a and pSmad1/5 expressions in LYVE1+ LECs were remarkably increased in the tendon of *FGFR3^Prox1* mice relative to controls (Supplementary Fig. 8b, c). Therefore, upregulated BMRP1a-pSmad1/5 signaling in LECs might be associated with

**Fig. 4 Ablation of *FGFR3* in LECs inhibits lymphatic formation and increases local inflammation levels during acquired HO development. a–e**
Representative confocal images of repaired Achilles tendon immunostained with LYVE1 (green) and DAPI (blue) in *Col2^tomato* and *FGFR3^{f/f}-Col2^tomato* mice at 4 and 8 weeks (bottom) after surgery and relative quantification (**c–e**). *n* = 4 per group. Dashed line boxes indicate tdTomato-labeled LYVE1^+ LECs (higher magnification, right). Scale bars, 100 µm (left); 20 µm (right). **f** mRNA levels of canonical LEC markers *LYVE1*, *Prox1*, and *PDPN* in the repaired Achilles tendon of *FGFR3^{f/f}* and *FGFR3^Col2* mice at 8 weeks after surgery. *n* = 3 per group. **g–i** Representative confocal images of repaired Achilles tendon immunostained with LYVE1 in *FGFR3^{f/f}* (**g**, *n* = 3) and *FGFR3^Prox1* mice (**h**, *n* = 4) at 8 weeks after surgery and lymphatic quantitative analysis. Dashed line boxes indicate LYVE1^+ lymphatics (higher magnification, right). Scale bars, 100 µm (left); 20 µm (right). **j, k** Representative ICG-NIR images of footpads immediately (0 min, top) and 24 h (bottom) after ICG administration in *FGFR3^{f/f}* (*n* = 8), *FGFR3^Col2* (*n* = 6), and *FGFR3^Prox1* mice (*n* = 6) 8 weeks after tenotomy and 24 h ICG clearance (**k**). **l–n** Representative confocal images of inflammatory macrophages immunostained with F4/80 (red), iNOS (green), and DAPI (blue) in the repaired Achilles tendon of *FGFR3^{f/f}* (*n* = 5), *FGFR3^Col2* (*n* = 4), and *FGFR3^Prox1* mice (*n* = 4) at 4 and 8 weeks (bottom) after surgery and relative quantification (**m, n**). Scale bars, 20 µm. White dashed lines indicate outlines of the tendon (**a, b, g**). All data represent mean ± SEM. *$P < 0.05$; **$P < 0.01$; ***$P < 0.001$ by unpaired two-tailed Student's *t*-test.

reduced lymphatic formation in the Achilles tendon post trauma in *FGFR3*-deficient mice.

To confirm whether the aggravated HO phenotype of *FGFR3* cKO mice is dependent on upregulated BMPR1a-pSmad1/5 signaling, *FGFR3^{f/f}; BMPR1a^{f/+}; Col2-CreER^{T2}* (*FGFR3^Col2; BMPR1a^{f/+}*) and *FGFR3^{f/f}; BMPR1a^{f/f}; Col2-CreER^{T2}* mice (*FGFR3^Col2;BMPR1a^{f/f}*) or *FGFR3^{f/f}; BMPR1a^{f/+}; Prox1-CreER^{T2}* (*FGFR3^Prox1;BMPR1a^{f/+}*) and *FGFR3^{f/f}; BMPR1a^{f/f}; Prox1-CreER^{T2}* mice (*FGFR3^Prox1;BMPR1a^{f/f}*) were generated to simultaneously delete *FGFR3* and *BMPR1a* in either Col2^+ lineage cells or LECs, respectively. *BMPR1a* cKO markedly decreased HO incidence in *FGFR3^Col2* mice. HO was not observed in the Achilles tendon of *FGFR3^Col2;BMPR1a^{f/f}* mice until 8 weeks post surgery (Supplementary Table 1). At 4 weeks after surgery, X-ray and µCT revealed reduced HO formation in *FGFR3* and *BMPR1a* double cKO mice relative to that in *FGFR3^Col2* mice (Supplementary Fig. 9a, b). H&E and SOFG staining demonstrated that ectopic cartilage was markedly reduced in *FGFR3^Col2;BMPR1a^{f/+}* mice compared with that in *FGFR3^Col2* mice and was largely absent in *FGFR3^Col2;BMPR1a^{f/f}* mice 4 weeks post surgery (Supplementary Fig. 9c, d). Immunostaining also demonstrated that the numbers of Sox9^+ chondrocytes and Osx^+ osteoblasts were markedly decreased in the repaired tendon of mice with double deletion of *FGFR3* and *BMPR1a* compared with those of *FGFR3^Col2* mice (Supplementary Fig. 9e, f). At 8 weeks post Achilles tenotomy, double deletion of *FGFR3* and *BMPR1a* in Col2^+ lineage cells led to decreased HO formation relative to *FGFR3^Col2* mice. Similarly, the heterotopic bone in the tendon of *FGFR3^Prox1* mice was significantly reduced following further deletion of *BMPR1a* in LECs at 8 weeks after surgery as indicated by µCT (Fig. 5h, i). Compared with the well-developed cancellous bone and marrow in the Achilles tendon of *FGFR3^Col2* mice 8 weeks post tenotomy, abundant ectopic cartilage instead of ossified bone was observed in *FGFR3^Col2;BMPR1a^{f/+}* mice as indicated by SOFG staining, which was further reduced in *FGFR3^Col2;BMPR1a^{f/f}* mice. Consistently, compared with the well-developed ectopic bone in *FGFR3^Prox1* mice, HO was significantly reduced in *FGFR3^Prox1;BMPR1a^{f/+}* mice, which was further attenuated in *FGFR3^Prox1;BMPR1a^{f/f}* mice (Fig. 5j, k). The number of OC^+ osteoblasts in the repaired tendon of either *FGFR3^Col2* or *FGFR3^Prox1* mice was significantly decreased following further deletion of *BMPR1a*, respectively (Fig. 5l, m). Moreover, immunostaining of LYVE1 demonstrated that the impaired lymphatic formation in the tendon of *FGFR3^Col2* or *FGFR3^Prox1* mice was alleviated by further deletion of *BMPR1a* at 8 weeks after tenotomy (Fig. 5n–p). The number of local F4/80^+iNOS^+ inflammatory macrophages was dramatically reduced in *FGFR3* and *BMPR1a* double mutants relative to that in *FGFR3^Col2* and *FGFR3^Prox1* mice, respectively (Fig. 5q). Altogether, our data suggest that *FGFR3* deficiency in LECs

suppresses local lymphatic formation and aggravates HO development in a BMPR1a-dependent manner.

**Local FGF9 treatment attenuates acquired HO formation in a lymphatic-dependent manner.** We next examined whether activating FGFR3 in LECs might attenuate HO development. FGF9 is known as a high-affinity ligand for FGFR[40]. Controlled release of growth factors using Matrigel (Corning), a widely used degradable hydrogel, has been used to induce tissue regeneration including angiogenesis[41], adipogenesis[42], and bone formation[43]. To examine the effect of FGF9 treatment on local lymphatic formation and HO development, we applied 0 (control), 0.01, 0.1, and 1 µg FGF9 with Matrigel on the injured Achilles tendon of wild-type (WT) mice immediately after Achilles tenotomy. The ectopic bone in the tendon was reduced in mice with 0.01, 0.1, and 1 µg FGF9 and Matrigel treatment relative to controls at 8 weeks post surgery (Fig. 6a). SOFG staining showed that well-developed HO was present in the tendon of control mice, which was mitigated in mice with FGF9 treatment (Fig. 6b, c). The numbers of Sox9^+ chondrocytes and OC^+ osteoblasts in HO lesions were significantly decreased in mice treated with FGF9 relative to control mice (Fig. 6d–g). Meanwhile, the numbers of LYVE1^+ LECs as well as ICG drainage were significantly increased in repaired Achilles tendons of mice locally treated with FGF9 (Fig. 6h–l). The number of local inflammatory macrophages was significantly decreased in WT mice with FGF9 treatment (Fig. 6m, n). However, no significant alteration of acquired HO formation revealed by µCT analysis or lymphatic vessels indicated by LYVE1 immunostaining was observed in Achilles tendons of *FGFR3^Prox1* mice with FGF9 treatment relative to controls at 8 weeks post surgery (Fig. 6o–s). In addition, significantly reduced HO formation was observed in WT mice with local application of VEGF-c in Matrigel 8 weeks post Achilles tenotomy (Supplementary Fig. 10a-d). Immunostaining revealed significantly increased numbers of LYVE1^+ LECs and reduced numbers of F4/80^+iNOS^+ inflammatory macrophages in the tendon of mice with VEGF-c administration relative to controls (Supplementary Fig. 10e-h). Taken together, we found that FGF9 treatment inhibits acquired HO development via FGFR3 activation-mediated local lymphatic formation.

**Discussion**
HO, the ectopic bone formed in soft tissues, has been known as a painful and recurrent disease that severely incapacitates the normal daily life of patients. We still have limited knowledge about the cellular and molecular mechanism of HO development. As a result, current therapy for acquired HO remains unsatisfied. One important question that needs to be explored is the involvement of specific mesenchymal progenitors during HO pathogenesis. Various markers have been used to label distinct

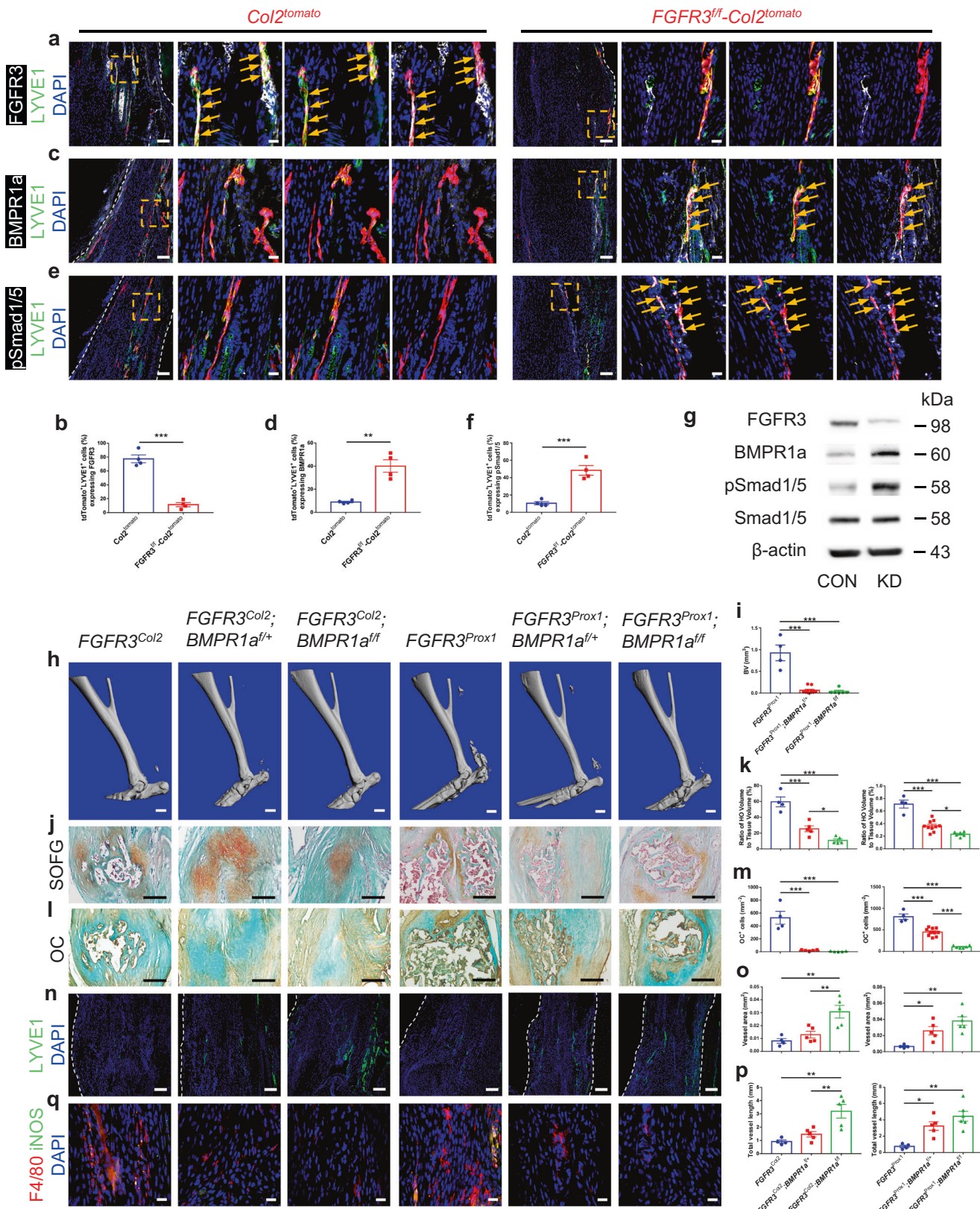

populations of mesenchymal stem cells (MSCs). Different MSC populations influence HO formation via diverse cellular and molecular mechanisms. Lineage tracing studies have demonstrated that Prx1+ and Gli1+ cells contribute to HO formation through their involvement in chondrogenesis and osteogenesis[9,10]. Nestin+ cells are recruited by overactivated TGF-β signaling for

chondrogenesis coupled with type H vessel formation during HO development. It is plausible that the heterogeneous Nestin+ cells provide both mesenchymal progenitors contributing to HO formation and endothelial progenitors involving in angiogenesis[3]. Previous study revealed that Col2+ cells encompass mesenchymal progenitors that give rise to chondrocytes, osteoblasts, adipocytes,

**Fig. 5 *FGFR3* deficiency in LECs aggravates acquired HO formation via upregulating BMPR1a-pSmad1/5 signaling. a–f** Representative confocal images of repaired Achilles tendon immunostained with FGFR3 (**a**), BMPR1a (**c**) or pSmad1/5 (**e**) (white), LYVE1 (green), and DAPI (blue) in *Col2^tomato* (left) and *FGFR3^f/f^-Col2^tomato* mice (right) at 8 weeks after surgery and relative quantitative analysis (**b**, **d**, **f**). n = 4 per group. Dashed line boxes indicate tdTomato-labeled LYVE1⁺ LECs (higher magnification with split channels, right). Scale bars, 100 μm (left); 20 μm (right). **g** Protein levels of FGFR3, BMPR1a, pSmad1/5, and Smad1/5 in mouse LEC cell line treated with *FGFR3*-targeted siRNA (KD) or control siRNA (CON). n = 3 per group. **h–m** Representative μCT (**h**), SOFG staining (**j**), and OC immunohistochemistry (**l**) images of ectopic bone in the Achilles tendon of *FGFR3^Col2* (n = 4), *FGFR3^Col2;BMPR1a^f/+* (n = 5), *FGFR3^Col2;BMPR1a^f/f* (n = 5), *FGFR3^Prox1* (n = 4), *FGFR3^Prox1;BMPR1a^f/+* (n = 10), and *FGFR3^Prox1;BMPR1a^f/f* mice (n = 6) at 8 weeks after surgery and relative quantitative analysis (**i**, **k**, **m**). Scale bars, 1 mm for μCT; 200 μm for SOFG and OC IHC. **n–q** Representative confocal images of Achilles tendon immunostained with LYVE1 (green) and DAPI (blue) (**n**) in *FGFR3^Col2* (n = 4), *FGFR3^Col2;BMPR1a^f/+* (n = 5), *FGFR3^Col2;BMPR1a^f/f* (n = 5), *FGFR3^Prox1* (n = 4), *FGFR3^Prox1;BMPR1a^f/+* (n = 5), and *FGFR3^Prox1;BMPR1a^f/f* mice (n = 6) and relative quantitative analysis (**o**, **p**) at 8 weeks after surgery. Scale bars, 100 μm. F4/80 (red), iNOS (green) and DAPI (blue) immunostained Achilles tendon sections (**q**). Scale bars, 20 μm. White dashed lines indicate outlines of the tendon (**a**, **c**, **e**, **n**). All data represent mean ± SEM. *P < 0.05; **P < 0.01; ***P < 0.001 by unpaired two-tailed Student's t-test (**b**, **d**, **f**) or by one-way ANOVA followed by a Tukey's multiple comparisons test (**i**, **k**, **m**, **o**, **p**).

and stromal cells in growing bones[30]. In the present study, we unexpectedly found that Col2⁺ lineage cells adopt LEC fate instead of differentiating into chondrocytes, osteoblasts, or vascular endothelial cells during HO development in the repaired Achilles tendon, which influences HO formation indirectly via affecting local lymphatic drainage and inflammatory levels. It is plausible that Col2⁺ mesenchymal progenitors are tissue-specific heterogeneous populations. Therefore, the distinct involvement of various mesenchymal progenitors in the development of HO needs to be precisely clarified by in vivo evidence including lineage tracing.

The origins of LECs remain controversial. The venous origin of LECs was first postulated by Florence Sabin in 1902[44] and was supported by lineage tracing of Prox1⁺ cells in developing mice in 2007[45]. Huntington and McClure proposed the mesenchymal origins of LECs in 1910[44]. It was reported that multipotent mesenchymal stem cells adopt LEC phenotypes in vitro and promote lymphatic regeneration in a mouse lymphedema model[46]. Though non-venous sources of LECs have been determined in the embryonic mouse heart[47] and skin[48], in vivo lineage tracing evidence supporting mesenchymal progenitor-derived lymphatics is still lacking. In this study, we showed that Col2⁺ cells acquire LEC fate in Achilles tendons after trauma. To our knowledge, this is the first study revealing a specific subpopulation of mesenchymal progenitors as a non-venous origin of LECs by in vivo lineage tracing. In the Achilles tendon without surgery, tdTomato-labeled Col2⁺ cells were observed in the peritendineum and adjacent connective tissues, which adopted LEC fate after 8 weeks of tamoxifen injection. Since no tdTomato-labeled Col2⁺ cells were observed in the blood vessels of repaired Achilles tendon, we speculate that Col2⁺ mesenchymal progenitors in the peritendineum and connective tissues adjacent to the Achilles tendon contribute to the lymphatic vessels in the repaired tendon. However, real-time in vivo evidence is still needed to confirm the local origin of LECs in the repaired Achilles tendon. Considering the tissue-specific features of lymphatics, it remains to be clarified whether Col2⁺ LEC progenitors are also involved in lymphatic formation in other organs such as heart, skin, and mesentery in pathophysiological setting. Cre recombinase driven by the *Col2a1* promoter has been widely used to target chondrocytes in mice[49,50]. Whether lymphatics derived from Col2⁺ LEC progenitors contribute to the phenotypes of these transgenic mice needs to be further clarified, especially in mouse models with surgical procedures such as bone fracture or traumatic osteoarthritis.

Previous HO-related studies have been mainly focused on the direct contribution of osteogenic cells including mesenchymal stem cells[3,9,10], fibroblasts[7] and vascular endothelial cells[8] to HO formation. The local microenvironment also plays a vital role in HO development. It was reported that overactive TGF-β signaling

drives HO progression by inducing type H vessel formation coupled with osteogenesis. Locally increased blood vessels transport more oxygen, nutrients, and minerals for HO development[3]. Besides, HO formation is promoted by local inflammatory cells including macrophages, mast cells, and lymphocytes, and is associated with systemic and local inflammatory cytokines including IL-1β, IL-6, and TNF-α[12–18]. Macrophages still persist in the Achilles tendon 3 weeks after injury[38]. TGF-β produced by macrophages has been found to contribute to HO development and TGF-β signaling remains activated in the osteogenesis stage of HO before a reduction till 15 weeks after injury. Therefore, it is plausible that chronic inflammatory components including macrophages play an important role for HO progression. The lymphatic system is essential for inflammatory resolution. OA progression is correlated with decreased local lymphatics[51] and promoting lymphatic formation has been found to relieve the development of chronic arthritis by reducing inflammation severity[24]. Our data indicate that HO development is alleviated by increasing lymphatic vessels with reduction of local inflammatory levels. Previous studies revealed that inflammatory cytokines including IFN-γ, TNF-α, and IL-1β suppress endothelial FGF signaling with reduced expression and activity of FGF signaling cascade[52]. In acquired HO specimens from traumatic patients, we found decreased lymphatics accompanied with downregulated FGFR3 expression in LECs and increased local inflammation during acquired HO progression, indicating that FGFR3 downregulation in LECs may act as an important event in HO development.

Previous studies demonstrated that FGFR3 is essential for lymphangiogenesis by regulating LEC proliferation as well as migration[28,29]. 9-cisRA-induced LEC proliferation was dependent on the upregulated FGFR3 expression in LECs, which promoted lymphatic vessel regeneration in various models of adult mice[29]. We found that LYVE1 expression in Col2-derived tomato⁺ cells was reduced in the repaired Achilles tendons of *FGFR3^f/f^-Col2-tomato* mice compared with *Col2^tomato* mice (Fig. 4a–c), which indicates that *FGFR3* deficiency inhibited LEC fate adoption of Col2⁺ cells in the tendon after injury. Previous studies reported that FGFR3 is an initial target of Prox1, which is known as a master regulator inducing lymphatic differentiation[28]. Therefore, it was speculated that FGFR3 may regulate LEC differentiation as well. Meanwhile, BMPR1a-pSmad1/5 signaling was found to act as the downstream of FGFR3 to regulate lymphangiogenesis in our present study. It was reported that BMP2-pSmad1/5 signaling inhibits lymphatic differentiation. BMP signaling was undetectable in developing LECs. Excess BMP2 signaling decreased LEC formation[39]. These findings indicate that *FGFR3* deficiency might also inhibit LEC differentiation via upregulated BMPR1a-pSmad1/5 signaling. Loss-of-function mutation in *FGFR3* has been reported to cause camptodactyly, tall stature, and hearing

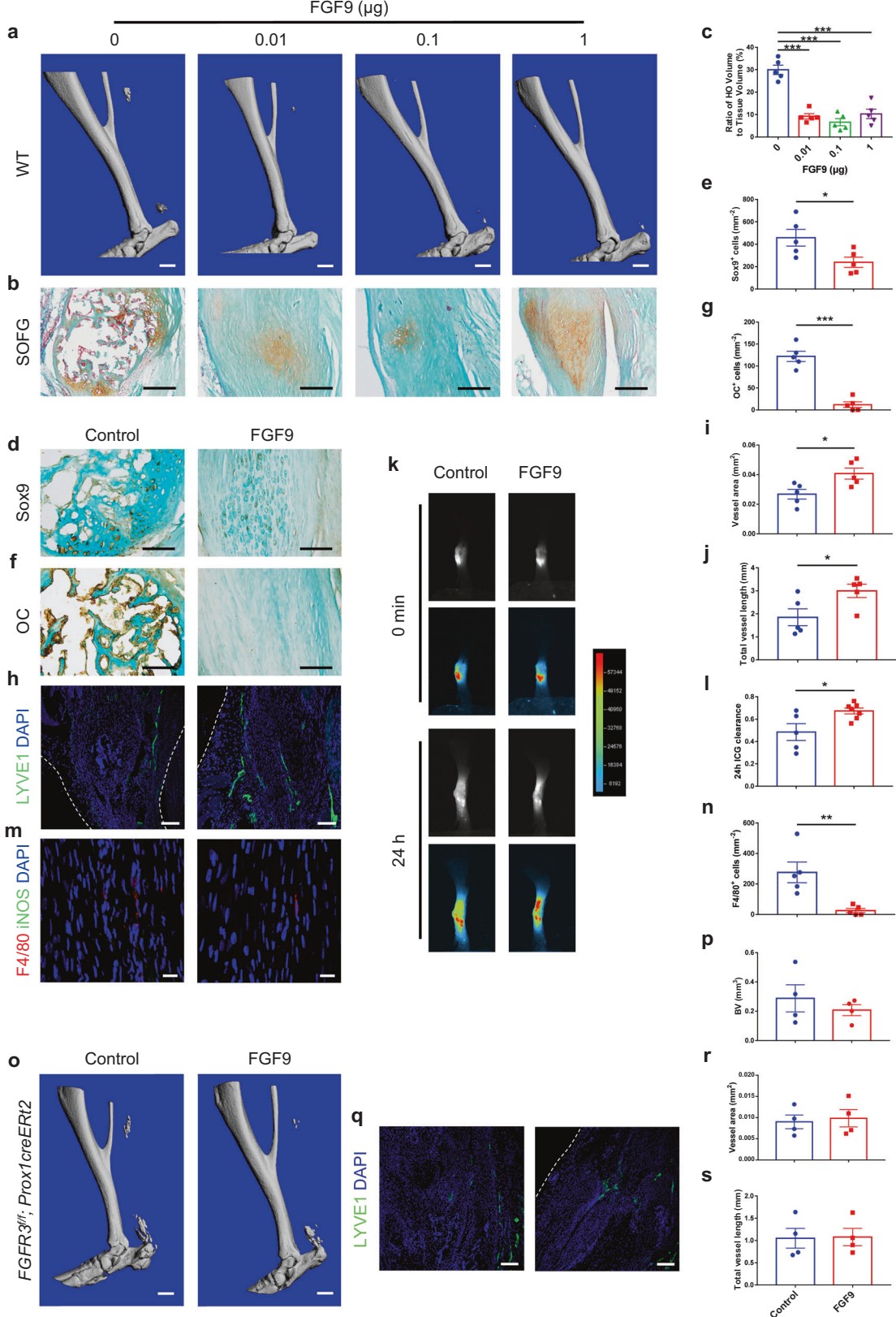

loss (CATSHL) syndrome[26]. Whether CATSHL patients have lymphatic malformation or a predisposition to acquired HO remains to be elucidated.

In summary, our present study revealed Col2+ lineage cells as an origin of LECs. *FGFR3* deletion in LECs decreased lymphatic formation after trauma via upregulated BMPR1a-pSmad1/5 signaling, which inhibited local lymphatic drainage and promoted inflammatory levels in the repaired tendon, leading to aggravated HO development (Fig. 7). Activation of FGFR3 with local application of FGF9 after surgery increased lymphatic vessels in the tendon

**Fig. 6 Local application of FGF9 inhibits acquired HO formation via activating FGFR3 in LECs. a–c** Representative μCT (**a**) and SOFG staining (**b**) images of heterotopic bone in the Achilles tendon of WT mice treated with 0 (control), 0.01, 0.1, or 1 μg FGF9 in Matrigel at 8 weeks after surgery and histomorphometry analysis (**c**). $n = 5$ per group. Scale bars, 1 mm for μCT; 200 μm for SOFG. **d–g** Representative Sox9 (**d**) and OC (**f**) IHC images of heterotopic bone in the Achilles tendon of WT mice treated with 0 (control) or 0.1 μg FGF9 in Matrigel at 8 weeks after surgery and relative quantification (**e**, **g**). $n = 5$ per group. Scale bars, 100 μm. **h–j** Representative confocal images of LYVE1 (green) and DAPI (blue) immunostained Achilles tendon sections (**h**) and lymphatic quantitative analysis (**i**, **j**). $n = 5$ per group. Scale bars, 100 μm. **k, l** Representative ICG-NIR images (**k**) of footpads immediately (0 min, top) and 24 h (bottom) after ICG administration in WT mice treated with FGF9 ($n = 7$) relative to controls ($n = 5$) at 8 weeks after tenotomy and 24 h ICG clearance (**l**). **m, n** Representative confocal images of Achilles tendon sections immunostained with F4/80 (red), iNOS (green), and DAPI (blue) (**m**) and relative quantitative analysis (**n**). Scale bars, 20 μm. **o, p** Representative μCT images (**o**) of heterotopic bone in the Achilles tendon of *FGFR3^Prx1* mice treated with FGF9 relative to controls at 8 weeks after surgery and quantitative analysis (**p**). $n = 4$ per group. Scale bars, 1 mm. **q–s** Representative confocal images of LYVE1 (green) and DAPI (blue) immunostained Achilles tendon sections (**q**) and lymphatic quantitative analysis (**r**, **s**). $n = 4$ per group. Scale bars, 100 μm. White dashed lines indicate outlines of the tendon (**h**, **q**). All data represent mean ± SEM. *$P < 0.05$; **$P < 0.01$; ***$P < 0.001$ by unpaired two-tailed Student's $t$-test (**e**, **g**, **i**, **j**, **l**, **n**, **p**, **r**, **s**) or by one-way ANOVA followed by a Tukey's multiple comparisons test (**c**).

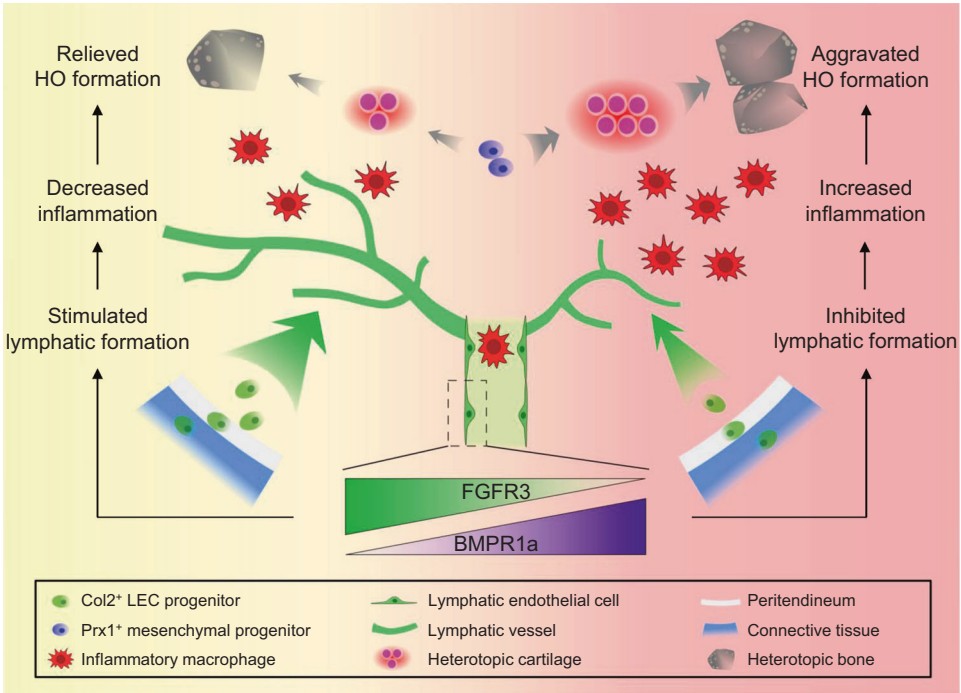

**Fig. 7 Schematic diagram describing the role of FGFR3 and lymphatics in acquired HO formation.** Col2[+] LEC progenitors in the peritendineum as well as adjacent connective tissues adopt LEC fate in the traumatically injured Achilles tendon at the onset of HO development. Downregulated FGFR3 expression in LECs associated with inflammatory milieu inhibits local lymphatic formation via upregulating BMPR1a signaling, which further increases local inflammatory levels and subsequently aggravates HO pathogenesis.

and inhibited HO formation, which was dependent on FGFR3 expression in LECs. Therefore, FGFR3 activation might be a prophylactic and therapeutic strategy for HO. In addition, our results also indicate that FGFR3 might be a therapeutic target for lymphatic dysfunction-related diseases such as human lymphedema.

## Methods

**Mice**. *FGFR3^flox/flox* (C3H/HeJ)[53], *BMPR1a^flox/flox* (C3H/HeJ)[54], *Col2a1-CreER^T2* (C3H/HeJ)[55], *Prox1-CreER^T2* (C57BL/6j)[56], *Prx1-CreER^T2* (C57BL/6j)[57], *Rosa26^tdTomato* (C57BL/6j) (Ai14, 007914, Jackson Laboratories, Bar Harbor, ME, USA), and *Rosa26^mTmG* (C57BL/6j) (007676, Jackson Laboratories, Bar Harbor, ME, USA) mice were all previously reported. *Prx1-CreER^T2*, *Prx1-CreER^T2*, *Rosa26^tdTomato*, and *Rosa26^mTmG* mice were backcrossed with C3H/HeJ mice for ten generations. *Col2a1-CreER^T2*, *Prx1-CreER^T2*, and *Prx1-CreER^T2* mice were mated with *FGFR3^flox/flox* mice to obtain *FGFR3^flox/flox*; *Col2-CreER^T2*, *FGFR3^flox/flox*; *Prox1-CreER^T2* and *FGFR3^flox/flox*; *Prx1-CreER^T2* mice. *FGFR3^flox/flox*; *Col2-CreER^T2* and *FGFR3^flox/flox*; *Prx1-CreER^T2* mice were mated with *BMPR1a^flox/flox* mice to obtain *FGFR3^flox/flox*; *BMPR1a^flox/+*; *Col2-CreER^T2*, *FGFR3^flox/flox*; *BMPR1a^flox/flox*; *Col2-CreER^T2*, *FGFR3^flox/flox*; *BMPR1a^flox/+*; *Prx1-CreER^T2* and *FGFR3^flox/flox*; *BMPR1a^flox/flox*; *Prx1-CreER^T2* mice. *Col2-CreER^T2* and *Prx1-CreER^T2* mice were mated with *Rosa26^tdTomato* mice to obtain *Col2-CreER^T2*;

*Rosa26^tdTomato* and *Prx1-CreER^T2*; *Rosa26^tdTomato* mice. *Col2-CreER^T2* mice were mated with *Rosa26^mTmG* mice to obtain *Col2-CreER^T2*; *Rosa26^mTmG* mice. *FGFR3^flox/flox*; *Col2-CreER^T2* and *FGFR3^flox/flox*; *Prx1-CreER^T2* mice were mated with *Rosa26^tdTomato* mice to obtain *FGFR3^flox/flox*; *Col2-CreER^T2*; *Rosa26^tdTomato* and *FGFR3^flox/flox*; *Prx1-CreER^T2*; *Rosa26^tdTomato* mice. For postnatal activation of CreER^T2, 10-week-old male mice were injected intraperitoneally with 100 mg/kg tamoxifen (T5648-5G, Sigma-Aldrich) in corn oil (C8267, Sigma-Aldrich) once a day for 5 consecutive days. To induce acquired HO formation, Achilles tenotomy was performed on male mice after tamoxifen injection according to the procedure. In brief, mice were anesthetized with 1% pentobarbital sodium. The skin was incised to expose the Achilles tendon. The Achilles tendon received 20 times of repeated clamping by the hemostatic forceps and then was cut by the scissors. Finally, the skin was closed with sutures.

All animals were maintained and handled with the approval of the Laboratory Animal Welfare and Ethics Committee of Daping Hospital (Chongqing, China). All mice were bred and maintained under SPF conditions with 12 h dark/light cycle, regular chow diet, 24 °C temperature, and 60% humidity.

**X-ray and μCT imaging**. X-ray images of HO specimens were obtained with the MX-20 Cabinet X-ray system (Faxitron X-Ray, Tucson, AZ, USA) according to the standard procedure. Undecalcified HO specimens were scanned by the vivaCT 40 μCT system (Scanco Medical, Brüttisellen, Switzerland) with the settings of 70 kV

and 113 mA. The threshold of 170 was applied for 3-D reconstructions and analysis of HO bone volume.

**Histological assessment and immunostaining analysis**. Fresh samples were fixed in 4% paraformaldehyde solution for 24 h, decalcified in 0.5 M EDTA solution for 2 weeks, and dehydrated in 30% sucrose solution for 24 h. Samples were embedded in OCT (Thermo) and sectioned at 10 μm intervals using a cryostat (Leica). Frozen sections were stained with hematoxylin and eosin (H&E) and Safranin O/Fast Green (SOFG). Histomorphometric analysis was conducted with the Osteomeasure Analysis System (Osteometrics) according to the standard procedure as previously described[58]. For immunofluorescence, frozen sections were blocked with goat serum for 30 min at 37 °C and then incubated at 4 °C overnight with primary antibodies including Sox9 (1:200; ab185230, Abcam), Runx2 (1:100; NBP1-77461, Novusbio), CD31 (1:100; FAB3628G, R&D Systems), LYVE1 (1:200; ab14917, Abcam/1:100; 14-0443-82, eBioscience), Prox1 (1:500; ab199359, Abcam), PDPN (1:200; ab11936, Abcam), VEGFR3 (1:100; AF743, R&D Systems), F4/80 (1:200; ab6640, Abcam), iNOS (1:50; ab15323, Abcam), α-SMA (1:200; A2547, Sigma-Aldrich), FGFR3 (1:100; BS90509, bioworld/1:50; sc-390423, Santa Cruz), BMPR1a (1:100; ab38560, Abcam), pSmad1/5 (1:100; 700047, Thermo). Secondary antibodies conjugated with fluorescence (1:500; Invitrogen/Abcam) were used at 37 °C for 1 h. Nuclei were counterstained by DAPI (1:1000; D8417, Sigma-Aldrich). Samples were imaged by LSM880NLO confocal microscope (Carl Zeiss). For immunohistochemistry, frozen sections were treated with 3% $H_2O_2$ at 37 °C for 15 min and then blocked with goat serum at 37 °C for 30 min. Primary antibodies including Sox9 (1:200; ab185230, Abcam), Osx (1:100; sc-22538, Santa Cruz), Runx2 (1:100; NBP1-77461, Novusbio), and OC (1:100; sc-365797, Santa Cruz) were used to incubate at 4 °C overnight. Appropriate biotinylated secondary antibodies were used followed by horseradish peroxidase-conjugated streptavidin-biotin staining to detect the immunoreactivity. Alcian blue was used for counterstaining. For human samples, about 50–100 serial sections of HO lesions were obtained from each of the five samples in each specimen. For animal studies, about 30–50 serial sections throughout repaired Achilles tendons were obtained from each sample. We conducted the quantitative analysis with 5–8 random visual fields per section of 6–10 sequential sections in each sample. For quantification of chondrocytes, osteoblasts, and macrophages, positive cells of relevant markers were counted manually with Image J software. For quantification of lymphatic vessels, LYVE1+ cells were assessed by AngioTool according to the standardized procedure as previously described[59]. All experiments were randomized and outcome assessments were obtained by investigators blinded to the allocation.

**Cell culture and *FGFR3* knockdown**. A murine lymphatic endothelial cell line (mLEC) established from benign lymphangiomas induced by Freund's adjuvant was used[60]. Murine LECs ($4 \times 10^5$/well) were seeded in 12-well plates and cultured in α-MEM medium (12571048, Gibco) containing 10% fetal bovine serum (FBS) (16140063, Gibco). For siRNA-mediated *FGFR3* knockdown, small interfering RNAs (siRNAs) for mouse *FGFR3* and for a negative control (RiboBio) were transfected with Lipofectamine RNAiMAX Transfection Reagent (13778100, Invitrogen) at 100 nM according to the manufacturer's instructions. Thirty-six hours after transfection, mLECs were collected in RIPA buffer for protein extraction.

**Isolation and culture of primary mouse tendon cells**. Repaired Achilles tendons of mice post surgery were obtained, cut into pieces, and digested with 3 mg/ml type I collagenase (17100-017, Gibco) in the incubator for 3 h. The cells were cultured in endothelial cell medium (ECM, 1001, ScienCell) for 7–10 d before trypsinized and reseeded on coverslips. After 3 d of culture, the cells were subjected to 4% paraformaldehyde fixation and immunofluorescent staining.

**Western blot analysis**. Achilles tendon tissues of mice post surgery were obtained and grinded with glass pestles in RIPA buffer on ice before ultrasonication for tissue protein extraction. Cell lysates were collected as described above. Samples were fractionated by 12% sodium dodecyl sulfate polyacrylamide gel electrophoresis (SDS-PAGE) and then transferred to polyvinylidene difluoride membranes (IPVH00010, Millipore). Immunoblotting was performed with primary antibodies against following proteins: Sox9 (1:2000; ab185230, Abcam), Runx2 (1:2000; NBP1-77461, Novusbio), FGFR3 (1:2000; BS90509, bioworld), BMPR1a (1:1000; ab38560, Abcam), pSmad1/5 (1:500; 700047, Thermo), Smad1/5 (1:1000; ab75273, Abcam), β-actin (1:5000; A8481, Sigma-Aldrich). Chemiluminescence was performed with SuperSignal West Dura Extended Duration Substrate (Thermo).

**Quantitative real-time PCR**. Total RNA was extracted from mouse Achilles tendon tissues with TRIzol reagent (15596-026, Invitrogen) and reverse transcription was conducted with PrimeScript RT reagent Kit with gDNA Eraser (Takara Biotechnology) according to the manufacturer's instructions. Quantitative analysis of gene expressions was conducted using SYBR Green (Takara Biotechnology) by a Mx3000P thermal cycler (Agilent Technologies). *Cyclophilin A* was used as the internal control. The primer sequences are shown in Supplementary Table 2.

**Local application of recombinant human FGF9 or VEGF-c in Matrigel**. Recombinant human FGF9 (100-23, Peprotech) was dissolved in double-distilled water at 0.004, 0.04, or 0.4 μg/μl and equally mixed with Matrigel of growth factor reduced type (356231, Corning) on ice at 0.002, 0.02, or 0.2 μg/μl, respectively. Double-distilled water was equally mixed with Matrigel on ice as the control group. Five microliters mixed solution of FGF9 (0, 0.01, 0.1, or 1 μg) and Matrigel was dripped on the Achilles tendon of male mice after surgery and the skin was gently sutured as described above after the mixture was gelled. Similarly, 5 μl mixed solution of recombinant human VEGF-c (100-20C, Peprotech) (0 or 0.1 μg) and Matrigel was applied locally in the Achilles tendon of male mice after tenotomy.

**Indocyanine green (ICG) near-infrared (NIR) lymphatic imaging**. Local lymphatic drainage in the repaired Achilles tendon was determined by ICG NIR lymphatic imaging as previously described[35]. Briefly, after the mice were anesthetized with 1% pentobarbital sodium, hair on the legs was removed by depilatory cream. ICG (I2633, Sigma) was dissolved in normal saline at 0.1 mg/ml and then 10 μl ICG solution was injected intradermally into the mouse footpad about 2 mm distal to the calcaneus. ICG imaging of mouse footpads and legs was collected by a Fusion FX Edge system (Vilber Lourmat). The signal intensity in the footpad was recorded immediately after ICG injection as the initial signal intensity. ICG imaging was collected again 24 h after ICG injection. Conditions including exposure time, focus, and position of the mouse hindlimbs need to be consistent for all imaging and overexposure needs to be avoided. The images were analyzed using Evolution-Capt v18.02 software. In brief, regions of interest (ROI) defining the injection site of the footpad was identified. ICG clearance was quantified as the percentage of reduced ICG signal intensity in the footpad 24 h post injection relative to the initial ROI signal intensity.

**Human HO specimen collection**. The study was approved by the Ethical and Protocol Review Committee of Daping Hospital (Chongqing, China). All experiments were performed according to approved guidelines. From January 2017 to October 2019, acquired HO specimens were obtained from patients undergoing surgeries in the Department of Trauma Surgery of Daping Hospital. HO specimens were collected from male patients who had previously sustained elbow fractures that were treated with internal fixation and returned for surgical treatment of acquired HO. All subjects with elbow fractures were previously healthy, non-smoking males aged between 25 and 45. Osteogenesis stage (3–6 months after fixation surgery) and maturation stage (12–18 months after fixation surgery) were defined by the period since their fixation surgery according to the reference[3]. All patients had no HO treatments including NSAIDs, local irradiation, or surgery during the course of our study. Informed consent had been obtained from all research participants in our study before their surgeries.

**Statistical analysis**. All data are presented as the mean ± SEM. The GraphPad PRISM 7 software (GraphPad Software, La Jolla, CA, USA) was used to analyze the statistics. Statistical methods are indicated in figure legends. $P < 0.05$ was considered statistically significant: *$P < 0.05$; **$P < 0.01$; ***$P < 0.001$. At least three independent biological replicates were conducted in each experiment to ensure reproducibility and replicates ($n$) were described in the figure legends.

**Reporting summary**. Further information on research design is available in the Nature Research Reporting Summary linked to this article.

## Data availability

All the data are available within the article and the source data file, and from the corresponding author upon reasonable request. There is no restriction for access to human samples.

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

## Acknowledgements

We thank all our colleagues of Dr. Chen's lab for technical support and valuable suggestions. This work is supported by grants from the National Key Research and Development Program of China (2018YFA0800802), National Natural Science Foundation of China (81830075, 81772306, 81991513, 81721001), Key Program of Innovation

Project of Military Medical Science (16CXZ016), and Innovative Research Team in University (IRT1216).

## Author contributions

D.Z., H.Q., Y.X., and Lin C. designed the study and analyzed the data. D.Z., J.H., X.S., H.C., Q.T., F.T.L., R.Z., W.J., Z.W., M.X., F.F.L., Liang C., M.L., and X.L. conducted the experiments. S.Z. provided the HO specimens of patients. X.D. and Lin C. contributed to reagents and materials. D.Z., Y.X., and Lin C. wrote the manuscript. S.H., J.Y., Z.N., M.J., N.S., L.Y., Y.Z., J.Q.F., D.C., H.Q., Y.X., and Lin C. reviewed and edited the manuscript.

## Competing interests

The authors declare no competing interests.
