## [Peer Review File · Nature Communications]

Reviewers' Comments:

Reviewer #1:

Remarks to the Author:

Comment:

In the manuscript "Targeting local lymphatics to ameliorate heterotopic ossification via FGFR3-BMPR1a pathway" Zhang et al. study mechanisms that underlie acquired heterotopic ossification (AHO). As a result of their study, the authors propose FGF signaling as a promising therapeutic target to treat the disorder.

Overall, the manuscript is preliminary and confusing and the presented data are not convincing. Especially, the lineage-tracing studies are of concern given the known leakiness of the genetic reporter used.

Other comments:

1. All the histology and immunofluorescence (IF) data are not properly labeled, making it difficult to evaluate the data. Also, the quality of IF images must be improved.
2. Why are the newly formed "lymphatics" CD31 negative (Fig. 2f)?
3. It has been reported that the R26tdTom reporter mouse shows Tamoxifen-independent recombination (Álvarez-Aznar, A et al., 2019). Consequently, the authors need to examine this possibility before concluding that LECs are of mesenchymal-origin.
4. It is a pity that the authors did not use Prox1 staining to demonstrate LEC identity.
5. The authors use LYVE1 as the only marker of lymphatics. However, LYVE1 is also expressed by resident macrophages. It would be more convincing to use a combination of PROX1 and LYVE1. To support their statement that COL2-positive cells can adopt lymphatic fate, they must sort these cells and profile their transcriptome.
6. The authors report that FGFR3 is involved in lymphatic migration and proliferation. However, the data do not exclude that FGFR3 is also involved in the differentiation from COL2-positive mesenchymal cells.
7. In supplementary Fig. 4, the authors claim that the reporter labeling indicates a high degree of lymphatic Fgfr3 deletion. However, reporter activity does not necessarily reflect gene deletion efficiency. To demonstrate this, the authors need to perform qPCR or immunoblot analyses.

Reference:

Álvarez-Aznar, A et al., Tamoxifen-independent recombination of reporter genes limits lineage tracing and mosaic analysis using CreERT2 lines. Transgenic Research 2019

Reviewer #2:

Remarks to the Author:

In this study, authors explored the role FGF signaling in acquired HO development. Their lineage tracing experiment showed that Col2+ cells adopted fate of lymphatic endothelial cells during HO development. FGFR3 cKO in Prox1-positive LECs increased HO formation. FGFR3 deficiency in LECs resulted in decreased local lymphatic formation with increased inflammatory levels. Local administration of FGF9 in Matrigel inhibited heterotopic bone formation. This study revealed Col2+ lineage cells as a novel origin of lymphatic endothelium in HO. This is an interesting novel finding. The experiments in general are well designed and executed. The data are convincing. I have the

following comments to improve the manuscript:

1. Heterotopic ossification is a very complex process involved in many factors. HO could occur only if all conditions are satisfied. This is why there are so many factors were identified in inhibition of HO. In the introduction, it did not describe the overall scheme of HO development and the potential role of FGF signaling in the process. For example, TGFbeta levels are significantly increased at both initial phase and late stage as well. And it is also critical for chondrogenesis and progression of HO. The information is missing in the Introduction.
2. There is no evidence to show the process of acquired heterotopic ossification is different from the other types of HO. AHO is already used for acute hematogenous osteomyelitis. AHO for acquired HO used here causes confusion in the field and literature.
3. Authors claim "we still have limited knowledge about the cellular and molecular mechanism of AHO development", but the manuscript did not even review the current understanding of four different stages of HO development and did not discuss their finding of FGF signaling in HO relative to the four stages of HO development.
4. BMP signaling is known to determine cell lineage fate. What is the function of BMP signaling in fate of lymphatic endothelial cells under normal physiology?
5. "Sustained high-level inflammation after trauma is related to impaired local lymphatic drainage in FGFR3-deficient mice, which may aggravate AHO development". Apparently, increase of local inflammatory levels subsequently in elevation of AHO is an indirect effect. Inflammatory is at early stage of HO, which promotes TGFbeta level for chondrogenesis. The authors should examine whether increase of TGFbeta activity for HO development.
6. The Diagram and Discussion should include overall outline of HO development and relative position of FGF signaling in LECs in HO.
7. The overall writing about HO and interpretation of the results need to be improved

Reviewer #3:

Remarks to the Author:

The manuscript by Zhang et al. introduces a compelling relationship between local lymphangiogenesis and acquired heterotopic ossification (AHO) in various sophisticated mouse models that underwent Achilles tenotomy. Specifically, the authors identified Col2+ resident progenitors of the peritendineum as a potential novel source of lymphatic endothelial cell (LEC) renewal post-tenotomy. The capacity for these progenitors to promote lymphangiogenesis post-tenotomy was directly associated with the severity of AHO development in a FGFR3 dependent manner. Conditional knockout (cKO) of FGFR3 in Col2+ progenitors and Prox1+ LECs led to increased AHO formation post-tenotomy, and this pathologic change was associated with an increase in BMPR1a and p-Smad1/5. Moreover, cKO of BMPR1a in these models reversed this phenotype. The authors propose that reduced lymphatic function promotes local inflammation that eventually dysregulates the FGFR3-BMPR1a signaling pathway leading to AHO formation, and thus targeting FGFR3 may promote lymphangiogenesis to ameliorate disease. While the manuscript presents a convincing story with data from both mice and humans to support their claims, there are concerns about some of the data and interpretation of some results that need to be addressed. There are also some minor concerns that the authors should consider.

Major Comments:

1. The current presentation of the images makes colocalization of markers difficult to assess. For example, the authors write, "Immunostaining revealed abundant expressions of canonical LEC markers LYVE1 and VEGFR3 in tdTomato labeled Col2+ lineage cells..." (Lines 200 - 202).

However, in the associated Figures 2g,h the colocalization of immunostain (green) and lineage trace (red) is questionable. As the authors point out, lymphatic vessels (LVs) were only present within the tendon after injury, so if Col2+ cells are truly the predominant progenitor for LECs in these circumstances, one would expect all (or most, dependent on tamoxifen efficiency) to be Col2-derived. Instead, in Figure 2h the VEGFR3 immunostaining appears to be mostly independent of the lineage traced cells besides a select few colocalized (yellow) cells. Moreover, for Figure 2g the presence of Col2-derived LYVE1+ cells is difficult to interpret as the lineage traced red fluorescence appears to only be present in the nuclei. How did the authors verify that these nuclei are specific to the LEC, and not the nuclei for the presumably directly adjacent lymphatic muscle cells? Is this lineage tracing mouse model expected to only show nuclear fluorescence, as it appears in other images (i.e. Figure 2h) to be non-specific to the nucleus and cytoplasm? An explanation for the lacking colocalization of many cells, or an alternative presentation of the fluorescence (i.e. split channels with a composite image) not just in Figure 2, but throughout the manuscript, is needed.

2. As presented, the conclusions of Figure 3 are misleading. The authors write, "Collectively, FGFR3 cKO in LECs tremendously promoted AHO development, which further supports that the aggravated AHO formation in FGFR3Col2 mice is strongly related to the disturbed LECs derived from Col2+ cells in the tendon after trauma" (Lines 293 – 296). While there may be a connection associated with the similar reduction in lymphangiogenesis when FGFR3 is deleted in both the proposed Col2-derived LEC progenitors and Prox1+ LECs, a direct mechanistic relationship between these two cells cannot be made as presented. The strongest conclusion that can be made is that there appears to be a relationship between lymphangiogenesis and AHO development represented in both models. Additional studies are needed to demonstrate that the associated cellular changes following FGFR3 deletion mediate the same lymphangiogenic disruption in both affected Col2+ progenitors and Prox1+ LECs. For instance, do Col2-derived Prox1+ LECs demonstrate continued disruption in the FGFR3-BMPRI1a pathway by protein expression in FGFR3f/f-Col2tomato compared to Col2tomato control mice? It may be possible that the reduced lymphangiogenesis in FGFR3Col2 animals functions by a similar, but unrelated mechanism compared to the FGFR3Prox1 model in which Col2+ progenitors are unable to proliferate and differentiate into LECs without FGFR3, and any lymphangiogenesis in FGFR3Col2 animals is due to ineffective deletion of FGFR3 in certain Col2+ progenitors. To make the claims as written, further experiments are needed to confirm that LECs derived from Col2+ progenitors in the FGFR3Col2 model indeed have disturbed FGFR3-BMPRI1a signaling leading to the reduced lymphangiogenesis via mechanisms similar to the FGFR3Prox1 construct.

3. The authors write, "Lineage tracing of FGFR3f/f; Prox1CreERT2; R26RtdTomato mice (FGFR3f/f-Prox1tomato) revealed that almost all LYVE1+ LECs in the tendon were labeled by tdTomato at 8 weeks post tenotomy, indicating a high efficiency of FGFR3 deletion in LECs in the repaired Achilles tendon" (Lines 277 – 280). Similar to Comment 1, the associated Supplementary Figure 4 seems to indicate very little colocalization between Prox1 lineage traced (red) and LYVE1 immunostained (green) cells. This finding brings into question the reliability of either the lineage tracing model or the immunostaining in these experiments, as Prox1 and LYVE1 should be colocalized as canonical LEC markers. Moreover, the title to Supplementary Figure 4, "Prox1+ lineage traced cells contribute to local LECs in the repaired tendon" is an inaccurate representation as LECs themselves are Prox1+.

4. LYVE1 is also a known marker of certain M2 polarized macrophages, especially surrounding smooth muscle cells to promote regulation of collagen content (PMID: 30054204). The authors demonstrate that under conditions where LYVE1 staining increases (depicted in the manuscript as equivalent to LECs, and used as a measure of LV area and length) that the number of M1 polarized macrophages decreases (Figures 4, Supp 5, 5, 6, Supp 7). Concurrent F4/80 or Prox1 staining ought to be performed with LYVE1 to confirm that LYVE1+ cells are truly representing LECs and the results are not confounded by LYVE1+ M2 polarized macrophages.

5. The near infrared indocyanine green (NIR-ICG) clearance as depicted in Figure 4j does not seem to match the quantified clearance results in Figure 4k. With the current images, it looks as if FGFR3Prox1 actually has greater or similar clearance compared to FGFR3f/f controls. A clarification of the analysis method depicted in the figures and more representative images in Figure 4j are

needed.

6. As presented, the Western blot in Figure 5e is questionable. It seems as if siFGFR3 #1 had relatively ineffective knockdown of FGFR3, but the pSmad1/5 levels appear increased similar to the other siFGFR3 lanes compared to the control. An explanation for this discrepancy is warranted. In addition, the authors write, "FGFR3 knockdown in mouse LEC line led to upregulated BMPR1a..." (Lines 434 – 435) however, as presented, it is difficult to see a noticeable increase in BMPR1a in the siFGFR3 conditions on the blot. Relative quantification of the protein levels would be helpful for interpreting these results.

7. Essential controls for the mouse models used in this study are missing. To validate accurate representation of the lineage tracing, Cre-negative and Cre-positive without tamoxifen induction (PMID 31641921) controls are necessary.

8. Clarification on the methods for immunostaining image analysis are needed. Were exact cell counts determined using an automated process or counted manually? How were the regions of interest for analysis determined in a representative and unbiased manner? Were the observers blinded to the conditions? "Image J software was used for quantifications. 5-8 independent images of 3-5 sequential sections in each sample were used for quantitative analysis" (Lines 733 – 735) is not sufficient to allow repeatability of this study.

9. For the human specimen collection, the methods note that, "HO specimens were collected from male patients who had previously sustained a femur or elbow fracture..." (Lines 786 – 787). In Supplementary Figure 5, are the human specimen data pooled to include both femur and elbow fractures? Is it expected for AHO formation in these two different conditions to behave similarly? What was the breakdown of subjects with elbow versus femur injuries that were assessed at the osteogenesis versus maturation stages?

10. Additional information on the mice used for the study is needed. In connection with Comment 9 in which it was noted for human subjects that only males were used, were both male and female mice used for this study? If so, were the sexes distributed evenly between the groups? Moreover, Jackson Laboratory stock numbers ought to be provided for the animals used in this study. For instance, according to Jackson Laboratories, the only Prx1-CreERT2 animal available is also tagged with a GFP (Prx1CreER-GFP; Stock No 029211). Was a different strain used for this study? If not, how was the Prx1-driven GFP fluorescence controlled in the analysis in Supplementary Figure 3, especially since the antibodies assessed were also labeled green? In addition, there are many Rosa26-tdTomato reporters offered by Jackson Laboratories, so which strain was used? This is also important to acknowledge given the differences in basal CreERT2 activity noted in the source for Comment 7.

Minor Comments:

1. There are potentially misleading comments that do not match the data as presented. For example, in reference to Supplementary Figure 5f, the authors write, "immunostaining revealed significantly increased numbers of F4/80+iNOS+ inflammatory macrophages in HO lesions at maturation stage relative to osteogenesis stage" (Lines 372 – 373). However, the figure demonstrates no significant difference in the % F4/80+iNOS+ cells relative to total F4/80+ cells between the two stages. The authors should ensure that all written explanations accurately depict the data as presented.

2. Figures are difficult to follow and require improved organization. Each figure should indicate the time point, injured versus uninjured, outlines of the tendon in all low-mag immunofluorescent images, and tissue being studied (relevant for Supp Fig. 2 and Supp Fig. 5).

3. The manuscript ought to be thoroughly proofread for grammar and typos.

We would like to submit the revised manuscript entitled "Targeting local lymphatics to ameliorate heterotopic ossification via FGFR3-BMP1a pathway" (NCOMMS-20-16420-T). We have addressed all reviewers' questions in the revised manuscript and provided 'point-to-point' replies. All changes were marked in red in the manuscript. We appreciate the opportunity allowing us to revise our manuscript.

Review 1

In the manuscript "Targeting local lymphatics to ameliorate heterotopic ossification via FGFR3-BMP1a pathway" Zhang et al. study mechanisms that underlie acquired heterotopic ossification (AHO). As a result of their study, the authors propose FGF signaling as a promising therapeutic target to treat the disorder.

Overall, the manuscript is preliminary and confusing and the presented data are not convincing. Especially, the lineage-tracing studies are of concern given the known leakiness of the genetic reporter used.

Response: Thank you very much for your valuable comments. We checked our manuscript according to your suggestions one by one. We carried out related experiments and hope our supplemental results or explanations might address your concerns.

Other comments:

1. All the histology and immunofluorescence (IF) data are not properly labeled, making it difficult to evaluate the data. Also, the quality of IF images must be improved.

Response: Thank you for your suggestion. For a better evaluation of images in the manuscript, we present the composite immunofluorescent images with split channels and/or modified the image intensity as shown in the Figures such as Fig2, Fig5, S Fig3, S Fig4, S Fig5, S Fig8, etc. For example, in Fig2g-I, reporter positive cells were shown in individual images with yellow arrows indicating Col2⁺ derived cells. Immunostainings of LYVE1, Prox1 and PDPN were also shown in individual images for a better evaluation of positive staining for LEC markers in Col2⁺ lineage cells.

2. Why are the newly formed "lymphatics" CD31 negative (Fig. 2f)?

Response: Thank you for your comment. We are sorry we mistakenly described the image. The tomato labeled cells in the injured tendon of Col2^{tomato} mice are CD31 positive (Fig2f), which indicates they are endothelial cells. Considering both vascular endothelial cells and lymphatic endothelial cells (LECs) are CD31 positive, we examined LEC marker LYVE1 and confirmed that these Col2-derived cells were co-stained with LYVE1 (Fig2g). Further *in vivo* and *in vitro* study with Col2^{mTmG} mice demonstrated that Col2-derived cells in the tendon post surgery were Prox1, LYVE1, PDPN and VEGFR3 positive (Fig2j-I, S Fig4a), which further confirms their LEC identity.

3. It has been reported that the R26tdTom reporter mouse shows Tamoxifen-independent recombination (Álvarez-Aznar, A et al., 2019). Consequently, the authors need to

examine this possibility before concluding that LECs are of mesenchymal-origin.

Response: We appreciate the reviewer's important suggestion. According to the referred paper¹, we tested tamoxifen-independent Cre recombination of reporter using *tomato* and *Col2^{tomato}* mice without tamoxifen induction as controls (S Fig3b). We did not find tomato positive cell in the repaired Achilles tendon of these control mice at 4 weeks post surgery, though co-staining of LYVE1 revealed that lymphatics were already formed in these tendons. Furthermore, as mentioned in this reference¹, *mTmG* reporter mice are more suitable for lineage tracing as they have lower recombination susceptibility. Therefore, we also used *Col2^{mTmG}* mice to further confirm the LEC identity of Col2-derived cells in the tendon after surgery *in vivo* and *in vitro*. GFP-labeled Col2⁺ lineage cells in the repaired tendon of *Col2^{mTmG}* mice were immunostained by different LEC markers including LYVE1, Prox1, VEGFR3 and PDPN at 4 weeks post tenotomy (Fig2j-l, S Fig4a). Similarly, without tamoxifen induction, GFP positive cells were not observed in the tendon of *Col2^{mTmG}* and *mTmG* controls at 4 weeks post tenotomy (S Fig3a, S Fig4b).

Reference:

1. Alvarez-Aznar, A., *et al.* Tamoxifen-independent recombination of reporter genes limits lineage tracing and mosaic analysis using CreER(T2) lines. *Transgenic Res* **29**, 53-68 (2020).

4. It is a pity that the authors did not use Prox1 staining to demonstrate LEC identity.

Response: We appreciate the suggestion. We performed Prox1 immunostaining in *Col2^{mTmG}* mice and confirmed that Col2-derived cells in the injured tendon were labeled by Prox1 (Fig2k). Moreover, we isolated primary cells in the repaired Achilles tendon of *Col2^{mTmG}* mice at 4 weeks post surgery and confirmed that Col2-derived GFP⁺ cells were stained by LEC markers including LYVE1, Prox1 and VEGFR3 (S Fig4a). These results demonstrated the LEC identity of Col2-derived cells in the tendon after surgery.

5. The authors use LYVE1 as the only marker of lymphatics. However, LYVE1 is also expressed by resident macrophages. It would be more convincing to use a combination of PROX1 and LYVE1. To support their statement that COL2-positive cells can adopt lymphatic fate, they must sort these cells and profile their transcriptome.

Response: We appreciate the important comment. We performed immunostainings of multiple LEC markers and confirmed that Col2-derived cells in the repaired tendon of *Col2^{mTmG}* mice were labeled by LYVE1, Prox1 and PDPN (Fig2j-l). Meanwhile, as mentioned in the manuscript (Lines 196-198), Col2-derived cells were stained by CD31 (Fig2f), which identified their fate of endothelial cells. Furthermore, we performed co-staining of LYVE1 and F4/80 in the repaired tendon of *Col2^{tomato}* mice as well as *FGFR3^{fl/fl}-Col2^{tomato}* mice, and found that Col2-derived cells in the tendon post surgery were stained by LYVE1 instead of F4/80 (S Fig5a,b). These results demonstrated that Col2⁺ cells adopted the fate of LECs rather than macrophages in the tendon after injury. Meanwhile, we tried to sort Col2-derived cells in the repaired tendon of *Col2^{mTmG}* mice.

However, the amount of primary GFP-labeled Col2⁺ lineage cells is too low to do sorting and transcriptome profiling. We also tried to amplify these primary GFP⁺ cells before sorting, but these GFP⁺ cells were amplified in a much slower rate than GFP⁻ cells *in vitro*, even though we used endothelial cell medium (ECM, ScienCell). Since *in vitro* evidence can help further confirm the LEC identity of Col2⁺ lineage cells in the repaired Achilles tendon, we isolated the primary cells in the repaired tendon of Col2^{tmG} mice and confirmed that GFP⁺ Col2-derived cells were immunostained by LEC markers including LYVE1, Prox1 and VEGFR3 (S Fig4a). We agree that the transcriptome profiling of Col2⁺ lineage cells is an important study, and we will carry it out using emerging new approaches in the future research. Thank you again for your important suggestion.

6. The authors report that FGFR3 is involved in lymphatic migration and proliferation. However, the data do not exclude that FGFR3 is also involved in the differentiation from COL2-positive mesenchymal cells.

Response: Thank you for your suggestion. Previous findings reported that FGFR3 is essential for lymphangiogenesis by regulating LEC proliferation as well as migration^{2,3}. It was also reported that FGFR3 is an initial target of Prox1, which is known as a master regulator inducing lymphatic differentiation². Therefore, it was speculated that FGFR3 may regulate LEC differentiation as well². Meanwhile, BMPR1a-pSmad1/5 signaling was found to act as the downstream of FGFR3 to regulate lymphangiogenesis in our present study. It was reported that BMP2-pSmad1/5 signaling inhibits lymphatic differentiation⁴, which indicates that *FGFR3* deficiency possibly inhibits LEC differentiation via upregulated BMPR1a-pSmad1/5 signaling. Furthermore, as shown in Fig 4a-c, LYVE1 immunostaining of Col2-derived tomato⁺ cells was reduced in the repaired Achilles tendons of *FGFR3^{fl/fl}-Col2^{tomato}* mice compared with *Col2^{tomato}* mice. Altogether, the above evidence from references and our present study indicate, as reviewer commented, that *FGFR3* deficiency may inhibit LEC fate adoption of Col2⁺ cells in the tendon after injury. We added this speculation in our discussion (Lines 709-719). However, it still remains to be confirmed whether and how FGFR3 regulates lymphatic differentiation of Col2⁺ cells. Thank you again for your kind reminding.

Reference:

2. Shin, J.W., *et al.* Prox1 promotes lineage-specific expression of fibroblast growth factor (FGF) receptor-3 in lymphatic endothelium: a role for FGF signaling in lymphangiogenesis. *Mol Biol Cell* **17**, 576-584 (2006).
3. Choi, I., *et al.* 9-cis retinoic acid promotes lymphangiogenesis and enhances lymphatic vessel regeneration: therapeutic implications of 9-cis retinoic acid for secondary lymphedema. *Circulation* **125**, 872-882 (2012).
4. Dunworth, W.P., *et al.* Bone morphogenetic protein 2 signaling negatively modulates lymphatic development in vertebrate embryos. *Circ Res* **114**, 56-66 (2014).

7. In supplementary Fig. 4, the authors claim that the reporter labeling indicates a high degree of lymphatic Fgfr3 deletion. However, reporter activity does not necessarily reflect gene deletion efficiency. To demonstrate this, the authors need to perform qPCR

or immunoblot analyses.

Response: We are sorry for our inaccurate description. *In situ* immunofluorescent co-staining of FGFR3 and LYVE1 was performed to examine the deletion efficiency of lymphatic FGFR3 in the repaired Achilles tendons of *FGFR3^{fl/fl}-Col2^{tomato}* and *FGFR3^{Prox1}* mice. As shown in Fig5a,b, S Fig8a, FGFR3 level was remarkably decreased in LYVE1⁺ LECs in both *FGFR3*-deficient mice.

Reference:

Álvarez-Aznar, A et al., Tamoxifen-independent recombination of reporter genes limits lineage tracing and mosaic analysis using CreERT2 lines. Transgenic Research 2019

Reviewer 2

In this study, authors explored the role FGF signaling in acquired HO development. Their lineage tracing experiment showed that Col2⁺ cells are adopted fate of lymphatic endothelial cells during HO development. FGFR3 cKO in Prox1-positive LECs increased HO formation. FGFR3 deficiency in LECs resulted in decreased local lymphatic formation with increased inflammatory levels. Local administration of FGF9 in Matrigel inhibited heterotopic bone formation. This study revealed Col2⁺ lineage cells as a novel origin of lymphatic endothelium in HO. This is an interesting novel finding. The experiments in general are well designed and executed. The data are convincing. I have the following comments to improve the manuscript:

1. Heterotopic ossification is a very complex process involved in many factors. HO could occur only if all conditions are satisfied. This is why there are so many factors were identified in inhibition of HO. In the introduction, it did not describe the overall scheme of HO development and the potential role of FGF signaling in the process. For example, TGFbeta levels are significantly increased at both initial phase and late stage as well. And it is also critical for chondrogenesis and progression of HO. The information is missing in the Introduction.

Response: Thank you very much for your suggestion. We added relevant descriptions in the introduction of the manuscript according to your comment. Acquired HO has been reported to develop through endochondral ossification involving stages of inflammation, chondrogenesis, osteogenesis and maturation. Immune cells and osteogenic progenitors are involved in HO development and various growth factors such as TGF-β have been reported to play important roles in regulating HO formation⁵ (Lines 61-64). TGF-β activity is indeed a very important factor regulating HO development, which was discussed in lines 645-649 (Nestin⁺ cells are recruited by overactivated TGF-β signaling for chondrogenesis coupled with type H vessel formation during HO development. It is plausible that the heterogeneous Nestin⁺ cells provide both mesenchymal progenitors contributing to HO formation and endothelial progenitors involving in angiogenesis⁵), 685-688 (It was reported that overactive TGF-β signaling drives HO progression by

inducing type H vessel formation coupled with osteogenesis. Locally increased blood vessels transport more oxygen, nutrients and minerals for HO development⁵ and 691-693 (TGF- β produced by macrophages has been found to contribute to HO development and TGF- β signaling remains activated in the osteogenesis stage of HO before a reduction till 15 weeks after injury⁵). The potential role of FGF signaling in HO was mentioned in lines 93-104 (Fibroblast growth factor (FGF) signaling plays an essential role in skeletal development⁶. Activation mutations of fibroblast growth factor receptor 3 (FGFR3) in human cause chondrodysplasia including achondroplasia, hypochondroplasia as well as thanatophoric dysplasia through inhibiting chondrocyte proliferation and differentiation⁷. Meanwhile, FGFR3 also plays a vital role in the regulation of lymphatic formation. FGFR3 is expressed in human and mouse lymphatic endothelial cells (LECs) and is essential for LEC proliferation and migration². 9-cis retinoic acid (9-cisRA) promotes LEC proliferation, migration and tube formation via activating FGF signaling. 9-cisRA-induced proliferation of LECs is coupled with increased FGFR3 expression, which is suppressed by soluble FGFR3 recombinant protein that sequesters FGF ligands³. All these findings suggest the possible involvement of FGFR3 in acquired HO development, although the accurate role and detailed underlying mechanisms remain to be clarified).

Reference:

2. Shin, J.W., *et al.* Prox1 promotes lineage-specific expression of fibroblast growth factor (FGF) receptor-3 in lymphatic endothelium: a role for FGF signaling in lymphangiogenesis. *Mol Biol Cell* **17**, 576-584 (2006).
3. Choi, I., *et al.* 9-cis retinoic acid promotes lymphangiogenesis and enhances lymphatic vessel regeneration: therapeutic implications of 9-cis retinoic acid for secondary lymphedema. *Circulation* **125**, 872-882 (2012).
5. Wang, X., *et al.* Inhibition of overactive TGF-beta attenuates progression of heterotopic ossification in mice. *Nat Commun* **9**, 551 (2018).
6. Xie, Y., Zhou, S., Chen, H., Du, X. & Chen, L. Recent research on the growth plate: Advances in fibroblast growth factor signaling in growth plate development and disorders. *J Mol Endocrinol* **53**, T11-34 (2014).
7. Qi, H., *et al.* FGFR3 induces degradation of BMP type I receptor to regulate skeletal development. *Biochim Biophys Acta* **1843**, 1237-1247 (2014).

2. There is no evidence to show the process of acquired heterotopic ossification is different from the other types of HO. AHO is already used for acute hematogenous osteomyelitis. AHO for acquired HO used here causes confusion in the field and literature.

Response: Thank you for your suggestion. We replaced 'AHO' with 'acquired HO' or 'HO' to avoid confusion.

3. Authors claim "we still have limited knowledge about the cellular and molecular mechanism of AHO development", but the manuscript did not even review the current understanding of four different stages of HO development and did not discuss their finding of FGF signaling in HO relative to the four stages of HO development.

Response: Thank you for your kind reminding. We added relevant information about the four stages of HO development in the introduction. Acquired HO has been reported to develop through endochondral ossification involving stages of inflammation, chondrogenesis, osteogenesis and maturation. Immune cells and osteogenic progenitors are involved in HO development and various growth factors such as TGF- β have been reported to play important roles in regulating HO formation⁵ (Lines 61-64). Previous studies revealed that inflammatory cytokines including IFN- γ , TNF- α and IL-1 β suppress endothelial FGF signaling with reduced expression and activity of FGF signaling cascade⁸. In HO samples from traumatic patients, we found decreased lymphatics accompanied with downregulated FGFR3 expression in LECs and increased local inflammation during HO progression, indicating that FGFR3 downregulation in LECs may act as an important event in HO development (Lines 699-704). Chronic inflammation remained in HO lesions during ectopic bone progression. Therefore, downregulated FGFR3 level in LECs might also be a prolonged factor that contributes to ectopic bone progression throughout later stages of HO. Thank you again for your kind suggestion.

Reference:

5. Wang, X., *et al.* Inhibition of overactive TGF-beta attenuates progression of heterotopic ossification in mice. *Nat Commun* **9**, 551 (2018).

8. Chen, P.Y., *et al.* FGF regulates TGF-beta signaling and endothelial-to-mesenchymal transition via control of let-7 miRNA expression. *Cell Rep* **2**, 1684-1696 (2012).

4. BMP signaling is known to determine cell lineage fate. What is the function of BMP signaling in fate of lymphatic endothelial cells under normal physiology?

Response: Thank you for your question. As described in the manuscript (Lines 716-718), BMP signaling was reported to play a negative role in LEC emergence during vertebrate development. BMP signaling is undetectable in developing LECs in embryos of both zebrafish and mouse models. Excess BMP2 signaling inhibits LEC formation via inducing miR-31 and miR-181a dependent on Smad4 and in turn inhibited Prox1 during lymphatic development⁴.

Reference:

4. Dunworth, W.P., *et al.* Bone morphogenetic protein 2 signaling negatively modulates lymphatic development in vertebrate embryos. *Circ Res* **114**, 56-66 (2014).

5. "Sustained high-level inflammation after trauma is related to impaired local lymphatic drainage in FGFR3-deficient mice, which may aggravate AHO development". Apparently, increase of local inflammatory levels subsequently in elevation of AHO is an indirect effect. Inflammation is at early stage of HO, which promotes TGFbeta level for chondrogenesis. The authors should examine whether increase of TGFbeta activity for HO development.

Response: Thank you for your good suggestion. According to the reference⁵, we conducted immunohistochemistry of pSmad2/3 in HO in the tendon and found that

pSmad2/3 levels in HO lesions were significantly increased in *FGFR3^{Col2}* and *FGFR3^{Prox1}* mice compared with *FGFR3^{fl/fl}* control group (data not shown). Therefore, upregulated TGF- β signaling may play an important role in the aggravation of acquired HO formation in *FGFR3*-deficient mice.

Reference:

5. Wang, X., *et al.* Inhibition of overactive TGF-beta attenuates progression of heterotopic ossification in mice. *Nat Commun* **9**, 551 (2018).

6. The Diagram and Discussion should include overall outline of HO development and relative position of FGF signaling in LECs in HO.

Response: We appreciate the reviewer's suggestion. We modified the diagram according to the comment. Meanwhile, FGF signaling in LECs during HO development was discussed in lines 699-704. Previous studies revealed that inflammatory cytokines including IFN- γ , TNF- α and IL-1 β suppress endothelial FGF signaling with reduced expression and activity of FGF signaling cascade⁸. In HO samples from traumatic patients, we found decreased lymphatics accompanied with downregulated FGFR3 expression in LECs and increased local inflammation during HO progression, indicating that FGFR3 downregulation in LECs may act as an important event in HO development (Lines 699-704). Chronic inflammation remained in HO lesions during ectopic bone progression. Therefore, downregulated FGFR3 level in LECs might also be a prolonged factor which contributes to ectopic bone progression throughout later stages of HO. Thank you again for your kind suggestion.

Reference:

8. Chen, P.Y., *et al.* FGF regulates TGF-beta signaling and endothelial-to-mesenchymal transition via control of let-7 miRNA expression. *Cell Rep* **2**, 1684-1696 (2012).

7. The overall writing about HO and interpretation of the results need to be improved

Response: Thank you for your suggestion. We thoroughly reviewed the whole manuscript and improved the interpretation of the results. For example, improved descriptions highlighted in red are shown in the results of our manuscript. Further discussions about the influence of FGFR3 on LEC differentiation were added in Lines 709-719.

Reviewer 3

The manuscript by Zhang et al. introduces a compelling relationship between local lymphangiogenesis and acquired heterotopic ossification (AHO) in various sophisticated mouse models that underwent Achilles tenotomy. Specifically, the authors identified Col2+ resident progenitors of the peritendineum as a potential novel source of lymphatic endothelial cell (LEC) renewal post-tenotomy. The capacity for these progenitors to promote lymphangiogenesis post-tenotomy was directly associated with the severity of AHO development in a FGFR3 dependent manner. Conditional knockout (cKO) of FGFR3 in Col2+

progenitors and Prox1+ LECs led to increased AHO formation post-tenotomy, and this pathologic change was associated with an increase in BMPR1a and p-Smad1/5. Moreover, cKO of BMPR1a in these models reversed this phenotype. The authors propose that reduced lymphatic function promotes local inflammation that eventually dysregulates the FGFR3-BMPR1a signaling pathway leading to AHO formation, and thus targeting FGFR3 may promote lymphangiogenesis to ameliorate disease. While the manuscript presents a convincing story with data from both mice and humans to support their claims, there are concerns about some of the data and interpretation of some results that need to be addressed. There are also some minor concerns that the authors should consider.

Major Comments:

1. The current presentation of the images makes colocalization of markers difficult to assess. For example, the authors write, "Immunostaining revealed abundant expressions of canonical LEC markers LYVE1 and VEGFR3 in tdTomato labeled Col2+ lineage cells..." (Lines 200 – 202). However, in the associated Figures 2g,h the colocalization of immunostain (green) and lineage trace (red) is questionable. As the authors point out, lymphatic vessels (LVs) were only present within the tendon after injury, so if Col2+ cells are truly the predominant progenitor for LECs in these circumstances, one would expect all (or most, dependent on tamoxifen efficiency) to be Col2-derived. Instead, in Figure 2h the VEGFR3 immunostaining appears to be mostly independent of the lineage traced cells besides a select few colocalized (yellow) cells. Moreover, for Figure 2g the presence of Col2-derived LYVE1+ cells is difficult to interpret as the lineage traced red fluorescence appears to only be present in the nuclei. How did the authors verify that these nuclei are specific to the LEC, and not the nuclei for the presumably directly adjacent lymphatic muscle cells? Is this lineage tracing mouse model expected to only show nuclear fluorescence, as it appears in other images (i.e. Figure 2h) to be non-specific to the nucleus and cytoplasm? An explanation for the lacking colocalization of many cells, or an alternative presentation of the fluorescence (i.e. split channels with a composite image) not just in Figure 2, but throughout the manuscript, is needed.

Response: Thank you very much for your important suggestions. We are sorry the merged images with no split channel in previous figures disturbed your assessment. The *Rosa26^{tdTomato}* mice here are Ai14 reporter mice (007914, Jackson Laboratories, Bar Harbor, ME, USA). We added images with split channels as shown in Fig2, Fig5, S Fig3, S Fig4, S Fig5, S Fig8, etc. for improving evaluation. Additionally, previous findings report multiple origins of LECs including venous endothelium and mesenchymal progenitors (Lines 659-665 of our MS). Therefore, Col2⁺ cells here are one important instead of sole origin of LECs in the repaired tendon, which contributed to about half of the LECs in the tendon post surgery. TdTomato-labeled Col2⁺ lineage cells contributed to approximately 53.3±3.4% and 56.6±2.8% of LYVE1⁺ LECs in the tendon of Col2^{tomato} mice at 4 and 8 weeks post surgery, respectively, which were reduced to 43.2±6.2% and 45.6±4.9% in FGFR3^{fl/fl}-Col2^{tomato} mice (Lines 369-372) (Fig4a,b,d). Yes, the quality of VEGFR3 staining in Fig2h is not satisfying. We replaced it with immunostainings of other LEC markers including LYVE1, Prox1 and PDPN in the repaired tendon of *Col2^{tmG}* mice (Fig2j-l).

Moreover, we also stained the primary cells isolated from repaired Achilles tendon of *Col2^{tomato}* mice with Prox1, LYVE1 and VEGFR3 *in vitro* (S Fig4a) and found that *Col2⁺* lineage cells in the tendon after surgery were labeled by these LEC markers.

2. As presented, the conclusions of Figure 3 are misleading. The authors write, "Collectively, FGFR3 cKO in LECs tremendously promoted AHO development, which further supports that the aggravated AHO formation in FGFR3Col2 mice is strongly related to the disturbed LECs derived from *Col2⁺* cells in the tendon after trauma" (Lines 293 – 296). While there may be a connection associated with the similar reduction in lymphangiogenesis when FGFR3 is deleted in both the proposed *Col2*-derived LEC progenitors and Prox1+ LECs, a direct mechanistic relationship between these two cells cannot be made as presented. The strongest conclusion that can be made is that there appears to be a relationship between lymphangiogenesis and AHO development represented in both models. Additional studies are needed to demonstrate that the associated cellular changes following FGFR3 deletion mediate the same lymphangiogenic disruption in both affected *Col2⁺* progenitors and Prox1+ LECs. For instance, do *Col2*-derived Prox1+ LECs demonstrate continued disruption in the FGFR3-BMPR1a pathway by protein expression in FGFR3f/f-*Col2*tomato compared to *Col2*tomato control mice? It may be possible that the reduced lymphangiogenesis in FGFR3Col2 animals functions by a similar, but unrelated mechanism compared to the FGFR3Prox1 model in which *Col2⁺* progenitors are unable to proliferate and differentiate into LECs without FGFR3, and any lymphangiogenesis in FGFR3Col2 animals is due to ineffective deletion of FGFR3 in certain *Col2⁺* progenitors. To make the claims as written, further experiments are needed to confirm that LECs derived from *Col2⁺* progenitors in the FGFR3Col2 model indeed have disturbed FGFR3-BMPR1a signaling leading to the reduced lymphangiogenesis via mechanisms similar to the FGFR3Prox1 construct.

Response: We appreciate your important suggestion. We carefully revised our conclusion according to your suggestion as follows: *FGFR3* cKO in LECs tremendously promoted acquired HO development, which indicates that there appears to be a causal relationship between dysregulated lymphangiogenesis and HO development represented in both *FGFR3^{Prox1}* and *FGFR3^{Col2}* mouse models (Lines 338-341). Meanwhile, immunostainings for FGFR3/BMPR1a/pSmad1/5 and LYVE1 were performed to evaluate the alterations of BMPR1a-pSmad1/5 signaling in LECs of repaired tendons in both *FGFR3^{Col2}* and *FGFR3^{Prox1}* constructs. FGFR3 level was downregulated, while the levels of BMPR1a and pSmad1/5 were upregulated in LECs of tendons post surgery in both *FGFR3*-deficient mouse models relative to controls (Fig5a-f, S Fig8a-c). Furthermore, *BMPR1a* deletion improved lymphatic formation and inhibited HO development in both *FGFR3*-deficient mice (Fig5h-q). Therefore, *FGFR3* deficiency in LECs leads to reduced lymphangiogenesis and aggravated HO development in both *FGFR3^{Col2}* and *FGFR3^{Prox1}* mice via upregulated BMPR1a-pSmad1/5 signaling. As shown in Fig5a,b, tomato⁺ *Col2*-derived cells that were not positively stained for FGFR3 were also labeled by LYVE1 in the repaired tendons of *FGFR3^{f/f}-Col2^{tomato}* mice relative to *Col2^{tomato}* mice, which

indicates that lymphangiogenesis from Col2⁺ progenitors with *FGFR3* deficiency was still present though decreased in repaired tendons of *FGFR3^{Col2}* mice, suggesting FGFR3 is an important but not indispensable receptor for lymphatic formation. Indeed, as the reviewer commented, the inhibition of lymphangiogenesis in *FGFR3^{Col2}* mice might also be a consequence of decreased lymphatic differentiation of Col2⁺ progenitors. As shown in Fig 4a-c, LYVE1 expression of Col2-derived tomato⁺ cells was reduced in the repaired Achilles tendons of *FGFR3^{ff}-Col2^{tomato}* mice compared with control group, which indicates that *FGFR3* deficiency inhibited LEC fate adoption of Col2⁺ cells in the tendon after injury. Previous studies reported that FGFR3 is an initial target of Prox1, which is known as a master regulator inducing lymphatic differentiation². Therefore, it was speculated that FGFR3 may regulate LEC differentiation as well². Meanwhile, BMPR1a-pSmad1/5 signaling was found to act as the downstream of FGFR3 to regulate lymphangiogenesis in our present study. It was reported that BMP2-pSmad1/5 signaling inhibits lymphatic differentiation⁴. All these data indicate that *FGFR3* deficiency possibly inhibits LEC differentiation via upregulated BMPR1a-pSmad1/5 signaling. Nevertheless, *FGFR3* deficiency might not completely block lymphatic differentiation since other molecules/signaling pathways are also involved in LEC differentiation such as VEGFc/VEGFR3 signaling pathway. Despite all the indications above, whether and how Col2⁺ progenitors without FGFR3, as reviewer suggested, might fail/decrease to differentiate/proliferate into LECs in the tendon after surgery is a very important and interesting topic that needs to be further investigated. Thank you very much for your thoughtful suggestion.

Reference:

2. Shin, J.W., *et al.* Prox1 promotes lineage-specific expression of fibroblast growth factor (FGF) receptor-3 in lymphatic endothelium: a role for FGF signaling in lymphangiogenesis. *Mol Biol Cell* **17**, 576-584 (2006).
4. Dunworth, W.P., *et al.* Bone morphogenetic protein 2 signaling negatively modulates lymphatic development in vertebrate embryos. *Circ Res* **114**, 56-66 (2014).
3. The authors write, "Lineage tracing of *FGFR3f/f; Prox1CreERT2; R26RtdTomato* mice (*FGFR3f/f-Prox1tomato*) revealed that almost all LYVE1+ LECs in the tendon were labeled by tdTomato at 8 weeks post tenotomy, indicating a high efficiency of *FGFR3* deletion in LECs in the repaired Achilles tendon" (Lines 277 – 280). Similar to Comment 1, the associated Supplementary Figure 4 seems to indicate very little colocalization between Prox1 lineage traced (red) and LYVE1 immunostained (green) cells. This finding brings into question the reliability of either the lineage tracing model or the immunostaining in these experiments, as Prox1 and LYVE1 should be colocalized as canonical LEC markers. Moreover, the title to Supplementary Figure 4, "Prox1+ lineage traced cells contribute to local LECs in the repaired tendon" is an inaccurate representation as LECs themselves are Prox1+.

Response: Thank you for your reminding. As shown in S Fig4, many tomato⁺ cells in the repaired tendon of *FGFR3^{ff}-Prox1^{tomato}* mice were not stained by LYVE1, though all LYVE1-stained LECs were tomato-positive. We speculate that Prox1 might also be

expressed by certain cells other than LECs in the tendon. To confirm this, further studies such as single-cell transcriptome sequencing of Prox1-traced cells may be needed. We also found that no Prox1⁺ cells contributed to HO formation, as no tomato⁺ cells were observed in HO lesions stained by SOX9 in the tendon of *FGFR3^{fl/fl}-Prox1^{tomato}* mice post surgery (data not shown). Therefore, *FGFR3* deficiency in LECs was regarded as the major cause of the aggravated HO formation with increased local inflammatory levels and decreased lymphangiogenesis in the repaired tendon of *FGFR3^{Prox1}* mice in this study. Considering reporter activity does not accurately reflect gene deletion efficiency, we replaced images in S Fig4 with FGFR3 and LYVE1 staining in the repaired tendon of *FGFR3^{Prox1}* mice (S Fig8a) to exhibit the *FGFR3* deletion efficiency in LECs in the tendon post surgery. Thank you very much for your comment.

4. LYVE1 is also a known marker of certain M2 polarized macrophages, especially surrounding smooth muscle cells to promote regulation of collagen content (PMID: 30054204). The authors demonstrate that under conditions where LYVE1 staining increases (depicted in the manuscript as equivalent to LECs, and used as a measure of LV area and length) that the number of M1 polarized macrophages decreases (Figures 4, Supp 5, 5, 6, Supp 7). Concurrent F4/80 or Prox1 staining ought to be performed with LYVE1 to confirm that LYVE1⁺ cells are truly representing LECs and the results are not confounded by LYVE1⁺ M2 polarized macrophages.

Response: Thank you for your suggestion. F4/80 and LYVE1 co-staining demonstrated that Col2-derived tomato⁺ cells were positively stained for LYVE1 instead of F4/80 in the repaired tendons of *Col2^{tomato}* mice and *FGFR3^{fl/fl}-Col2^{tomato}* mice (S Fig5a,b). LYVE1⁺ cells in the tendon were not stained by F4/80, which indicates that these LYVE1⁺ cells were LECs instead of macrophages. Meanwhile, our *in vivo* and *in vitro* evidence showed that Col2-derived cells in the repaired tendon of *Col2^{mtmG}* mice were immunostained by Prox1, VEGFR3, PDPN and LYVE1 (Fig2j-l, S Fig4a). CD31 staining also indicated endothelial identity of Col2-derived cells in the tendon post injury (Fig2f). Altogether, these evidence revealed that Col2-derived LYVE1⁺ cells in the repaired Achilles tendon are LECs instead of macrophages.

5. The near infrared indocyanine green (NIR-ICG) clearance as depicted in Figure 4j does not seem to match the quantified clearance results in Figure 4k. With the current images, it looks as if *FGFR3^{Prox1}* actually has greater or similar clearance compared to *FGFR3^{fl/fl}* controls. A clarification of the analysis method depicted in the figures and more representative images in Figure 4j are needed.

Response: Thank you for your suggestion. We replaced the previous NIR-ICG image of *FGFR3^{Prox1}* mice with a more representative image (Fig4j). A clarification of NIR-ICG analysis was explained in the methods and materials as follows: The signal intensity in the footpad was recorded immediately after ICG injection as the initial signal intensity. ICG imaging was collected again 24 hours after ICG injection. Conditions including exposure time, focus and position of the mouse hindlimbs need to be consistent for all imaging

and overexposure needs to be avoided. The images were analyzed using Evolution-Capt v18.02 software. In brief, regions of interest (ROI) defining the injection site of the footpad was identified. ICG clearance was quantified as the percentage of reduced ICG signal intensity in the footpad 24 hours post injection relative to the initial ROI signal intensity (Lines 843-850).

6. As presented, the Western blot in Figure 5e is questionable. It seems as if siFGFR3 #1 had relatively ineffective knockdown of FGFR3, but the pSmad1/5 levels appear increased similar to the other siFGFR3 lanes compared to the control. An explanation for this discrepancy is warranted. In addition, the authors write, "FGFR3 knockdown in mouse LEC line led to upregulated BMPR1a..." (Lines 434 – 435) however, as presented, it is difficult to see a noticeable increase in BMPR1a in the siFGFR3 conditions on the blot. Relative quantification of the protein levels would be helpful for interpreting these results.

Response: Thank you for your suggestion. The seemingly inconsistency of the western blot result might be due to the variable *FGFR3*-knockdown efficiency among these three siRNAs. To obtain a more stable and efficient knockdown of *FGFR3*, siFGFR3#1, #2 and #3 were pooled together for a combined *FGFR3* knockdown in mLEC line. As shown in Fig5g, FGFR3 level was evidently knocked down and BMPR1a/pSmad1/5 levels were remarkably upregulated.

7. Essential controls for the mouse models used in this study are missing. To validate accurate representation of the lineage tracing, Cre-negative and Cre-positive without tamoxifen induction (PMID 31641921) controls are necessary.

Response: Thank you for your suggestion. According to the referred paper¹, tamoxifen-independent Cre recombination of reporter mice was tested using *tomato* and *Col2^{tomato}* mice without tamoxifen induction as controls (S Fig3b). We did not find tomato positive cell in the repaired Achilles tendon of these control mice at 4 weeks post surgery, though co-staining of LYVE1 revealed that lymphatics were already formed in these tendons. Furthermore, as mentioned in this reference¹, *mTmG* reporter mice are more suitable for lineage tracing as they have lower recombination susceptibility. Therefore, to obtain more accurate lineage tracing, we also used *Col2^{mTmG}* mice to further confirm the LEC identity of Col2-derived cells in the tendon after surgery *in vivo* and *in vitro*. GFP-labeled Col2⁺ lineage cells in the repaired tendon of *Col2^{mTmG}* mice were immunostained by different LEC markers including LYVE1, Prox1, VEGFR3 and PDPN at 4 weeks post tenotomy (Fig2j-l, S Fig4a). Similarly, without tamoxifen induction, GFP positive cells were not observed in the tendon of *Col2^{mTmG}* and *mTmG* controls at 4 weeks post tenotomy (S Fig3a, S Fig4b). Thank you again for your valuable suggestion.

Reference:

1. Alvarez-Aznar, A., *et al.* Tamoxifen-independent recombination of reporter genes limits lineage tracing and mosaic analysis using CreER(T2) lines. *Transgenic Res* **29**, 53-68 (2020).

8. Clarification on the methods for immunostaining image analysis are needed. Were exact cell counts determined using an automated process or counted manually? How were the regions of interest for analysis determined in a representative and unbiased manner? Were the observers blinded to the conditions? "Image J software was used for quantifications. 5-8 independent images of 3-5 sequential sections in each sample were used for quantitative analysis" (Lines 733 – 735) is not sufficient to allow repeatability of this study.

Response: Thank you for your suggestion. For human samples, about 50-100 serial sections of HO lesions were obtained from each of 5 samples in each specimen. For animal studies, about 30-50 serial sections throughout repaired Achilles tendons were obtained from each sample. We conducted the quantitative analysis with 5-8 random visual fields per section of 6-10 sequential sections in each sample (We mistakenly regard 'sequential sections' as sequential slides in the previous manuscript and there were 2-3 sections on each slide). For quantification of chondrocytes, osteoblasts and macrophages, positive cells of relevant markers were counted manually with Image J software. For quantification of lymphatic vessels, LYVE1⁺ cells were assessed by AngioTool according to the standardized procedure as previously described⁹. All experiments were randomized and outcome assessment were obtained by investigators blinded to the allocation (Lines 787-796).

Reference:

9. Zudaire, E., Gambardella, L., Kurcz, C. & Vermeren, S. A computational tool for quantitative analysis of vascular networks. *PLoS One* **6**, e27385 (2011).

9. For the human specimen collection, the methods note that, "HO specimens were collected from male patients who had previously sustained a femur or elbow fracture..." (Lines 786 – 787). In Supplementary Figure 5, are the human specimen data pooled to include both femur and elbow fractures? Is it expected for AHO formation in these two different conditions to behave similarly? What was the breakdown of subjects with elbow versus femur injuries that were assessed at the osteogenesis versus maturation stages?

Response: Thank you for your suggestion. We excluded the data of the subject with femur fracture to avoid inconsistency in the study. All subjects with elbow fractures were previously healthy, nonsmoking males aged between 25 and 45. Osteogenesis stage (3-6 months after fixation surgery) and maturation stage (12-18 months after fixation surgery) were defined by the period since their fixation surgery according to the reference⁵. All patients had no HO treatments including NSAIDs, local irradiation or surgery during the course of our study (Lines 857-861).

Reference:

5. Wang, X., *et al.* Inhibition of overactive TGF-beta attenuates progression of heterotopic ossification in mice. *Nat Commun* **9**, 551 (2018).

10. Additional information on the mice used for the study is needed. In connection with

Comment 9 in which it was noted for human subjects that only males were used, were both male and female mice used for this study? If so, were the sexes distributed evenly between the groups? Moreover, Jackson Laboratory stock numbers ought to be provided for the animals used in this study. For instance, according to Jackson Laboratories, the only Prx1-CreERT2 animal available is also tagged with a GFP (Prx1CreER-GFP; Stock No 029211). Was a different strain used for this study? If not, how was the Prx1-driven GFP fluorescence controlled in the analysis in Supplementary Figure 3, especially since the antibodies assessed were also labeled green? In addition, there are many Rosa26-tdTomato reporters offered by Jackson Laboratories, so which strain was used? This is also important to acknowledge given the differences in basal CreERT2 activity noted in the source for Comment 7.

Response: Thank you for your suggestion. We are sorry for the missed information of mice in the manuscript. All mice used in the study were male. The *Prx1-CreER^{T2}* mice used in the study were from Dr. Malcolm Logan, National Institute for Medical Research, London, UK and were previously reported^{10,11}. The *Rosa26^{tdTomato}* mice used in this study were *Ai14* mice (007914, Jackson Laboratories).

Reference:

10. Hasson, P., Del Buono, J. & Logan, M.P. Tbx5 is dispensable for forelimb outgrowth. *Development* **134**, 85-92 (2007).

11. Jin, H., *et al.* Anti-DKK1 antibody promotes bone fracture healing through activation of beta-catenin signaling. *Bone* **71**, 63-75 (2015).

Minor Comments:

1. There are potentially misleading comments that do not match the data as presented. For example, in reference to Supplementary Figure 5f, the authors write, "immunostaining revealed significantly increased numbers of F4/80+iNOS+ inflammatory macrophages in HO lesions at maturation stage relative to osteogenesis stage" (Lines 372 – 373). However, the figure demonstrates no significant difference in the % F4/80+iNOS+ cells relative to total F4/80+ cells between the two stages. The authors should ensure that all written explanations accurately depict the data as presented.

Response: Thank you for your comment. We revised the description and replaced it with the following sentence: "immunostaining revealed a significantly increased number of F4/80⁺ macrophages in HO lesions at maturation stage relative to osteogenesis stage and the percentage of F4/80⁺iNOS⁺ inflammatory macrophages was also increased though without significant difference".

2. Figures are difficult to follow and require improved organization. Each figure should indicate the time point, injured versus uninjured, outlines of the tendon in all low-mag immunofluorescent images, and tissue being studied (relevant for Supp Fig. 2 and Supp Fig. 5).

Response: Thank you for your suggestion. We re-organized the figures and added

essential information in the figures.

3. The manuscript ought to be thoroughly proofread for grammar and typos.

Response: Thank you for your suggestion. We thoroughly proofread the manuscript to avoid mistakes. We will further improve the overall writing with the help of language editing company once this MS is accepted.

Reviewers' Comments:

Reviewer #1:

Remarks to the Author:

Zhang and colleagues have submitted a revision of their manuscript that contains new experiments and clarifications. The revised paper is improved and addresses several of my previous concerns, while others remain. Given the rather positive comments by the other two referees, the manuscript seems acceptable.

Reviewer #2:

Remarks to the Author:

My questions are adequately addressed.

Reviewer #3:

Remarks to the Author:

The authors have done a brilliant job responding to all the comments from the reviewers. There are no remaining concerns.

We would like to submit the revised manuscript entitled "Targeting local lymphatics to ameliorate heterotopic ossification via FGFR3-BMP1a pathway" (NCOMMS-20-16420B). Here we provide 'point-to-point' replies to the reviewers. All changes were marked in red in the manuscript. We appreciate the opportunity allowing us to revise our manuscript.

Reviewer #1 (Remarks to the Author):

Zhang and colleagues have submitted a revision of their manuscript that contains new experiments and clarifications. The revised paper is improved and addresses several of my previous concerns, while others remain. Given the rather positive comments by the other two referees, the manuscript seems acceptable.

Response: Thank you very much for your valuable comments.

Reviewer #2 (Remarks to the Author):

My questions are adequately addressed.

Response: Thank you very much.

Reviewer #3 (Remarks to the Author):

The authors have done a brilliant job responding to all the comments from the reviewers.

There are no remaining concerns.

Response: Thank you very much.